# PARP: Prune, Adjust and Re-Prune
# for Self-Supervised Speech Recognition

**Cheng-I Jeff Lai[1], Yang Zhang[2]**∗, **Alexander H. Liu[1]**∗, **Shiyu Chang[2, 4]**∗
**Yi-Lun Liao[1], Yung-Sung Chuang[1, 3], Kaizhi Qian[2], Sameer Khurana[1]**
**David Cox[2], James Glass[1]**
[1]MIT CSAIL, [2]MIT-IBM Watson AI Lab, [3]National Taiwan University, [4]UC Santa Barbara

clai24@mit.edu

## Abstract

Self-supervised speech representation learning (speech SSL) has demonstrated the benefit of scale in learning rich representations for Automatic Speech Recognition (ASR) with limited paired data, such as wav2vec 2.0. We investigate the existence of sparse subnetworks in pre-trained speech SSL models that achieve even better low-resource ASR results. However, directly applying widely adopted pruning methods such as the Lottery Ticket Hypothesis (LTH) is suboptimal in the computational cost needed. Moreover, we show that the discovered subnetworks yield minimal performance gain compared to the original dense network.

We present Prune-Adjust-Re-Prune (PARP), which discovers and finetunes subnetworks for much better performance, while only requiring a *single* downstream ASR finetuning run. PARP is inspired by our surprising observation that subnetworks pruned for pre-training tasks need merely a slight adjustment to achieve a sizeable performance boost in downstream ASR tasks. Extensive experiments on low-resource ASR verify (1) sparse subnetworks exist in mono-lingual/multi-lingual pre-trained speech SSL, and (2) the computational advantage and performance gain of PARP over baseline pruning methods.

In particular, on the 10min Librispeech split without LM decoding, PARP discovers subnetworks from wav2vec 2.0 with an absolute 10.9%/12.6% WER decrease compared to the full model. We further demonstrate the effectiveness of PARP via: cross-lingual pruning without any phone recognition degradation, the discovery of a multi-lingual subnetwork for 10 spoken languages in 1 finetuning run, and its applicability to pre-trained BERT/XLNet for natural language tasks[1].

## 1 Introduction

For many low-resource spoken languages in the world, collecting large-scale transcribed corpora is very costly and sometimes infeasible. Inspired by efforts such as the IARPA BABEL program, Automatic Speech Recognition (ASR) trained without sufficient transcribed speech data has been a critical yet challenging research agenda in speech processing [31, 33, 42, 32, 21]. Recently, Self-Supervised Speech Representation Learning (speech SSL) has emerged as a promising pathway toward solving low-resource ASR [84, 25, 110, 6, 29, 127, 55, 27]. Speech SSL involves pre-training a speech representation module on large-scale *unlabelled* data with a self-supervised learning objective, followed by finetuning on a small amount of supervised transcriptions. Many recent studies have demonstrated the empirical successes of speech SSL on low-resource English and multi-lingual ASR, matching systems trained on fully-supervised settings [6, 29, 127, 4, 126]. Prior research attempts, however, focus on pre-training objectives [84, 25, 110, 72, 57, 74, 71, 73, 55, 22, 27, 17, 129], scaling up speech representation modules [5, 6, 53], pre-training data selections [108, 54, 107, 111, 78], or

---

∗Equal contribution.

[1]Project webpage: https://people.csail.mit.edu/clai24/parp/

applications of pre-trained speech representations [26, 62, 94, 28, 63, 29, 76, 120, 64, 117, 112, 44, 4, 86, 59, 66, 2, 56, 102, 15, 30, 20]. In this work, we aim to develop an orthogonal approach that is complementary to these existing speech SSL studies, that achieves 1) lower architectural complexity and 2) higher performance (lower WER) under the same low-resource ASR settings.

Neural network pruning [65, 51, 49, 69], as well as the more recently proposed Lottery Ticket Hypothesis (LTH) [39], provide a potential solution that accomplishes both objectives. According to LTH, there exists sparse subnetworks that can achieve the same or *even better* accuracy than the original dense network. Such phenomena have been successfully observed in various domains: Natural Language Processing (NLP) [123, 19, 88, 80], Computer Vision (CV) [18, 45], and many others. All finding sparse subnetworks with comparable or better performance than the dense network. Given the lack of similar studies on pruning self-supervised ASR, we intend to fill this gap by finding sparse subnetworks *within a pre-trained* speech SSL that can achieve superior performance to the full pre-trained model on downstream ASR tasks.

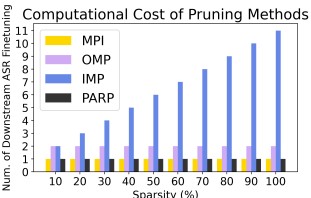

However, directly applying widely-adopted pruning methods, such as One-Shot Magnitude Pruning (OMP) and Iterative Magnitude Pruning (IMP) [49, 39], to pre-trained speech SSL suffers from two challenges. First, adopting these methods in the conventional pruning framework is extremely time-consuming for SOTA speech SSL models. OMP and IMP involve more than one round of finetuning on downstream tasks (c.f. Figure 1), and finetuning for ASR is time-consuming and computationally demanding[2]. The second challenge is that we do not observe *any* performance improvement of the subnetworks over the original dense network with OMP or IMP. Figure 3 shows the WER under low-resource scenarios of the subnetworks identified by OMP (purple line) and IMP (blue dashed line) at different sparsity levels. None of the sparsity levels achieves a visible drop in WER compared to the zero sparsity case, corresponding to the original dense network. These two challenges have prompted us to ask – do there exist sparse subnetworks within pre-trained speech SSL with improved performance on low-resource ASR? How can we discover them efficiently in a *single* downstream finetuning run?

Figure 1: Number of ASR finetuning iterations needed (y-axis) versus target sparsities (x-axis) for *each* downstream task/language. Cross-referencing Figure 3 indicates that IMP requires linearly more compute to match the performance (either sparsity/WER) of PARP.

We propose a magnitude-based unstructured pruning method [41, 11], termed Prune-Adjust-Re-Prune (PARP), for discovering sparse subnetworks within pre-trained speech SSL. PARP consists of the following two steps:

1. Directly prune the SSL pre-trained model at target sparsity, and obtain an initial subnetwork and an initial pruning mask.
2. Finetune the initial subnetwork on target downstream task/language. During finetuning, zero out the pruned weights specified by the pruning mask, but allow the weights be updated by gradient descent during backpropogation. After a few number of model updates, re-prune the updated subnetwork at target sparsity again.

Step 1 provides an initial subnetwork that is agnostic to the downstream task, and Step 2 makes learnable adjustments by reviving pruned out weights. A formal and generalized description and its extension are introduced in Section 3. Different from pruning methods in [49, 39], PARP allows pruned-out weights to be revived during finetuning. Although such a high-level idea was introduced in [48], we provide an alternative insight: despite its flexibility, Step 2 only makes **minimal adjustment** to the initial subnetwork, and obtaining a good initial subnetwork in Step 1 is the key. We empirically show in Section 3 that *any* task-agnostic subnetwork surprisingly provides a good basis for Step 2, suggesting that the initial subnetwork can be cheaply obtained either from a readily available task/language or directly pruning the pre-trained SSL model itself. In addition, this observation allows us to perform cross-lingual pruning (mask transfer) experiments, where the initial subnetwork is obtained via a different language other than the target language.

**Our Contributions.** We conduct extensive PARP and baseline (OMP and IMP) pruning experiments on low-resource ASR with mono-lingual (pre-trained wav2vec 2.0 [6]) and cross-lingual (pre-trained XLSR-53 [29]) transfer. PARP finds significantly superior speech SSL subnetworks for low-resource

---

[2]Standard wav2vec 2.0 finetuning setup [6] on any Librispeech/Libri-light splits requires at least 50∼100 V100 hours, which is more than 50 times the computation cost for finetuning a pre-trained BERT on GLUE [106].

ASR, while only requiring a single pass of downstream ASR finetuning. Due to its simplicity, `PARP` adds minimal computation overhead to existing SSL downstream finetuning.

- We show that sparse subnetworks exist in pre-trained speech SSL when finetuned for low-resource ASR. In addition, `PARP` achieves superior results to `OMP` and `IMP` across all sparsities, amount of finetuning supervision, pre-trained model scale, and downstream spoken languages. Specifically, on Librispeech 10min without LM decoding, `PARP` discovers subnetworks from wav2vec 2.0 with an absolute 10.9%/12.6% WER decrease compared to the full model, without modifying the finetuning hyper-parameters or objective (Section 4.1).

- Ablation studies on demonstrating the importance of `PARP`'s initial subnetwork (Section 4.2).

- `PARP` minimizes phone recognition error increases in cross-lingual mask transfer, where a subnetwork pruned for ASR in one spoken language is adapted for ASR in another language (Section 4.3). `PARP` can also be applied to efficient multi-lingual subnetwork discovery for 10 spoken languages (Section 4.4).

- Last but not least, we demonstrate `PARP`'s effectiveness on pre-trained BERT/XLNet, mitigating the cross-task performance degradation reported in BERT-Ticket [19] (Section 4.5).

**Significance.** Findings of this work not only complement and advance current and future speech SSL for low-resource ASR, but also provide new insights for the rich body of pruning work.

## 2 Preliminaries

### 2.1 Problem Formulation

Consider the low-resource ASR problem, where there is only a small transcribed training set $(x, y) \in \mathcal{D}_l$. Here $x$ represents input audio, and $y$ represents output transcription. Subscript $l \in \{1, 2, \cdots\}$ represents the downstream spoken language identity. Because of the small dataset size, empirical risk minimization generally does not yield good results. Speech SSL instead assumes there is a much larger unannotated dataset $x \in \mathcal{D}_0$. SSL pre-trains a neural network $f(x; \theta)$, where $\theta \in \mathcal{R}^d$ represents the network parameters and $d$ represents the number of parameters, on some self-supervised objective, and obtains the pre-trained weights $\theta_0$. $f(x; \theta_0)$ is then finetuned on downstream ASR tasks specified by a downstream loss $\mathcal{L}_l(\theta)$, such as CTC, and evaluated on target dataset $\mathcal{D}_l$.

Our goal is to discover a subnetwork that minimizes downstream ASR WER on $\mathcal{D}_l$. Formally, denote $m \in \{0, 1\}^d$, as a binary pruning mask for the pre-trained weights $\theta_0$, and $\theta^l$ as the finetuned weights on $\mathcal{D}_l$. The ideal pruning method should learn $(m, \theta^l)$, such that the subnetwork $f(x; m \odot \theta^l)$ (where $\odot$ is element-wise product) achieves minimal finetuning $\mathcal{L}_l(\theta)$ loss on $\mathcal{D}_l$.

### 2.2 Pruning Targets and Settings

We adopted pre-trained speech SSL `wav2vec2` and `xlsr` for the pre-trained initialization $\theta_0$.

**wav2vec 2.0** We took wav2vec 2.0 base (`wav2vec2-base`) and large (`wav2vec2-large`) pre-trained on Librispeech 960 hours [6]. During finetuning, a task specific linear layer is added on top of `wav2vec2` and jointly finetuned with CTC loss. More details can be found in Appendix 8.

**XLSR-53** (`xlsr`) shares the same architecture, pre-training and finetuning objectives as `wav2vec2-large`. `xlsr` is pre-trained on 53 languages sampled from CommonVoice, BABEL, and Multilingual LibriSpeech, totaling for 56k hours of multi-lingual speech data.

We consider three settings where `wav2vec2` and `xlsr` are used as the basis for low-resource ASR:

**LSR: Low-Resource English ASR.** Mono-lingual pre-training and finetuning – an English pre-trained speech SSL such as `wav2vec2` is finetuned for low-resource English ASR.

**H2L: High-to-Low Resource Transfer for Multi-lingual ASR.** Mono-lingual pre-training and multi-lingual finetuning – a speech SSL pre-trained on a high-resource language such as English is finetuned for low-resource multi-lingual ASR.

**CSR: Cross-lingual Transfer for Multi-lingual ASR.** Multi-lingual pre-training and finetuning – a cross-lingual pretrained speech SSL such as `xlsr` is finetuned for low-resource multi-lingual ASR.

### 2.3 Subnetwork Discovery in Pre-trained SSL

One obvious solution to the aforementioned problem in Section 2.1 is to directly apply pruning with rewinding to $\theta_0$, which has been successfully applied to pre-trained BERT [19] and SimCLR [18].

All pruning methods, including our proposed `PARP`, are based on Unstructured Magnitude Pruning (`UMP`) [39, 41], where weights of the lowest magnitudes are pruned out regardless of the network structure to meet the target sparsity level. We introduce four pruning baselines below, and we also provide results with Random Pruning (`RP`) [39, 41, 19], where weights in $\theta_0$ are randomly eliminated.

**Task-Aware Subnetwork Discovery** is pruning with target dataset $D_l$ seen in advance, including One-Shot Magnitude Pruning (`OMP`) and Iterative Magnitude Pruning (`IMP`). `OMP` is summarized as:

1. Finetune pretrained weights $\theta_0$ on target dataset $\mathcal{D}_l$ to get the finetuned weights $\theta^l$.
2. Apply `UMP` on $\theta^l$ and retrieve pruning mask $m$.

`IMP` breaks down the above subnetwork discovery phase into multiple iterations – in our case multiple downstream ASR finetunings. Each iteration itself is an `OMP` with a fraction of the target sparsity pruned. We follow the `IMP` implementation described in BERT-Ticket [19], where each iteration prunes out 10% of the *remaining* weights. The main bottleneck for `OMP` and `IMP` is the computational cost, since multiple rounds of finetunings are required for subnetwork discovery.

**Task-Agnostic Subnetwork Discovery** refers to pruning without having seen $D_l$ nor $l$ in advance. One instance is applying `UMP` directly on $\theta_0$ without any downstream finetuning to retrieve $m$, referred to as Magnitude Pruning at Pre-trained Initailizations (`MPI`). Another case is pruning weights finetuned for a different language $t$, *i.e.* applying `UMP` on $\theta^t$ for the target language $l$; in our study, we refer to this as cross-lingual mask transfer. While these approaches do not require target task finetuning, the discovered subnetworks generally have worse performance than those from `OMP` or `IMP`.

The above methods are only for subnetwork discovery via applying pruning mask $m$ on $\theta_0$. The discovered subnetwork $f(x; m \odot \theta_0)$ needs another downstream finetuning to recover the pruning loss[3], *i.e.* finetune $f(x; m \odot \theta_0)$ on $D_l$.

## 3 Method

In this section, we highlight our proposed pruning method, `PARP` (Section 3.1), its underlying intuition (Section 3.2), and an extension termed `PARP-P` (Section 3.3).

### 3.1 Algorithm

We formally describe `PARP` with the notations from Section 2. A visual overview of `PARP` is Figure 8.

---

**Algorithm 1** Prune-Adjust-Re-Prune (PARP) to target sparsity $s$

---

1: Assume there are $N$ model updates in target task/language $l$'s downstream finetuning.
2: Take a pre-trained SSL $f(x; \theta_0)$ model. Apply task-agnostic subnetwork discovery, such as `MPI`[4], at target sparsity $s$ to obtain initial subnetwork $f(x; m_0 \odot \theta_0)$. Set $m = m_0$ and variable $n_1 = 0$ .
3: **repeat**
4:     Zero-out masked-out weights in $\theta_{n1}$ given by $m$. Lift up $m$ such that whole $\theta_{n1}$ is updatable.
5:     Train $f(x; \theta_{n1})$ for $n$ model updates and obtain $f(x; \theta_{n2})$.
6:     Apply `UMP` on $f(x; \theta_{n2})$ and adjust $m$ accordingly. The adjusted subnetwork is $f(x; m \odot \theta_{n2})$. Set variable $n_1 = n_2$.
7: **until** total model updates reach $N$.
8: Return finetuned subnetwork $f(x; m \odot \theta_N)$.

---

Empirically, we found the choice of $n$ has little impact. In contrast to `OMP/IMP/MPI`, `PARP` allows the pruned-out weights to take gradient descent updates. A side benefit of `PARP` is it jointly discovers and finetunes subnetwork in a single pass, instead of two or more in `OMP` and `IMP`.

### 3.2 Obtaining and Adjusting the Initial Subnetwork

`PARP` achieves superior or comparable pruning results as task-aware subnetwork discovery, while inducing similar computational cost as task-agnostic subnetwork discovery. How does it get the best of both worlds? The key is the discovered subnetworks from task-aware and task-agnostic prunings have high, non-trivial overlaps in LSR, H2L, and CSR. We first define Intersection over Union (`IOU`) for quantifying subnetworks' (represented by their pruning masks $m^a$ and $m^b$) similarity:

$$\text{IOU}(m^a, m^b) \triangleq \frac{|(m^a = 1) \cap (m^b = 1)|}{|(m^a = 1) \cup (m^b = 1)|} \tag{1}$$

---

[3]This step is referred to as subnetwork finetuning/re-training in the pruning literature [75, 93, 11].
[4]By default, `MPI` is used for obtaining the initial subnetwork for `PARP` and `PARP-P` unless specified otherwise.

Take H2L and CSR for instance, Figure 2 visualizes language pairs' `OMP` pruning mask `IOUs` on `wav2vec2` and `xlsr`. Observe the high overlaps across all pairs, but also the high `IOUs` with the `MPI` masks (second to last row). We generalize these observations to the following:

> **Observation 1** *For any sparsity, any amount of finetuning supervision, any pre-training model scale, and any downstream spoken languages, the non-zero ASR pruning masks obtained from task-agnostic subnetwork discovery has high IOUs with those obtained from task-aware subnetwork discovery.*

Observation 1 suggests that *any* task-agnostic subnetwork could sufficiently be a good initial subnetwork in `PARP` due to the high similarities. In the same instance for H2L and CSR, we could either take `MPI` on `wav2vec2` and `xlsr`, or take `OMP` on a different spoken language as the initial subnetworks. Similarly in LSR, we take `MPI` on `wav2vec2` as the initial subnetwork. The underlying message is – the initial subnetwork can be obtained cheaply, without target task finetuning.

Now, because of the high similarity, the initial subnetwork (represented by its pruning mask $m_0$) needed merely a slight adjustment for the target downstream task. While there are techniques such as dynamic mask adjustment [48], important weights pruning [79], and deep rewiring [10], we provide an even simpler alternative suited for our setting. Instead of permanently removing the masked-out weights from the computation graph, `PARP` merely zeroes them out. Weights that are important for the downstream task (the "important weights") should emerge with gradient updates; those that are relatively irrelevant should decrease in magnitude, and thus be zero-outed at the end. Doing so circumvents the need of straight-through estimation or additional sparsity loss, see Table 1 of [97].

### 3.3 PARP-Progressive (`PARP-P`)

An extension to `PARP` is `PARP-P`, where the second P stands for Progressive. In `PARP-P`, the initial subnetwork starts at a lower sparsity, and progressively prune up to the target sparsity $s$ in Step 2. The intuition is that despite Observation 1, *not any* subnetwork can be a good initial subnetwork, such as those obtained from RP, or those obtained at very high sparsities in `MPI`/`OMP`/`IMP`. We show later that `PARP-P` is especially effective in higher sparsity regions, e.g. 90% for LSR. Note that `PARP-P` has the same computational cost as `PARP`, and the only difference is the initial starting sparsity in Step 1.

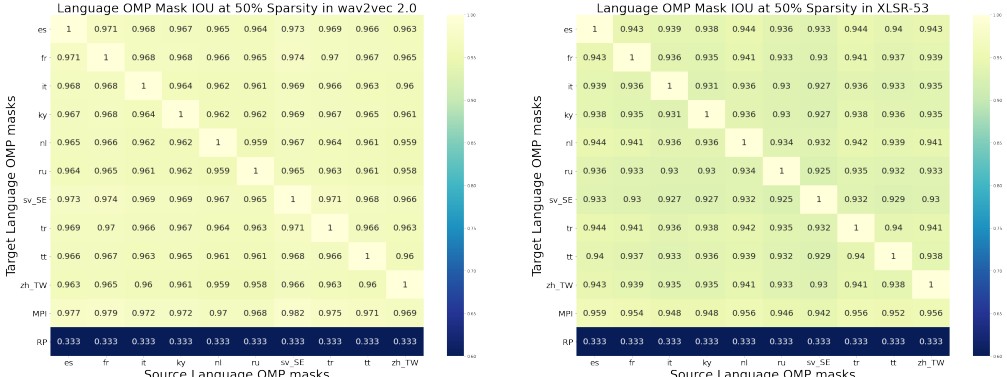

Figure 2: `IOUs` over all spoken language pairs' `OMP` pruning masks on finetuned `wav2vec2` and `xlsr`. Second to last row is the `IOUs` between `OMP` masks and the `MPI` masks from pre-trained `wav2vec2` and `xlsr`. Here, we show the `IOUs` at 50% sparsity, and the rest can be found in Appendix 11. Surprisingly at any sparsities, there is a high, non-trivial (c.f. RP in the last row), similarity (>90%) between all spoken language `OMP` masks, as well as with the `MPI` masks. Language IDs are in Appendix 9.

## 4 Experiments and Analysis

### 4.1 Comparing `PARP`, `OMP`, and `IMP` on LSR, H2L, and CSR

Our experimental setup can be found in Appendix 9. We first investigate the existence of sparse subnetworks in speech SSL. Figure 3 shows the pruning results on LSR. Observe that subnetworks discovered by `PARP` and `PARP-P` can achieve 60~80% sparsities with minimal degradation to the full models. The gap between `PARP` and other pruning methods also widens as sparsities increase. For instance, Table 1 compares `PARP` and `PARP-P` with `OMP` and `IMP` at 90% sparsity, and `PARP-P` has a 40% absolute WER reduction. In addition, observe the WER reduction with `PARP` in the low sparsity

regions on the 10min split in Figure 3. The same effect is not seen with `OMP`, `IMP`, nor `MPI`. Table 2 compares the subnetworks discovered by `PARP` with the full `wav2vec2` and prior work on LSR under the same setting[5]. Surprisingly, the discovered subnetwork attains an absolute 10.9%/12.6% WER reduction over the full `wav2vec2-large`. We hypothesize that the performance gains are attributed to pruning out generic, unnecessary weights while preserving important weights, which facilitates training convergence. In other words, `PARP` provides additional regularization effects to downstream finetuning. We also examined the effectiveness of `IMP` with different rewinding starting points as studied in [40, 93], and found rewinding initializations bear minimal effect on downstream ASR. Full rewinding details are in Appendix 10.

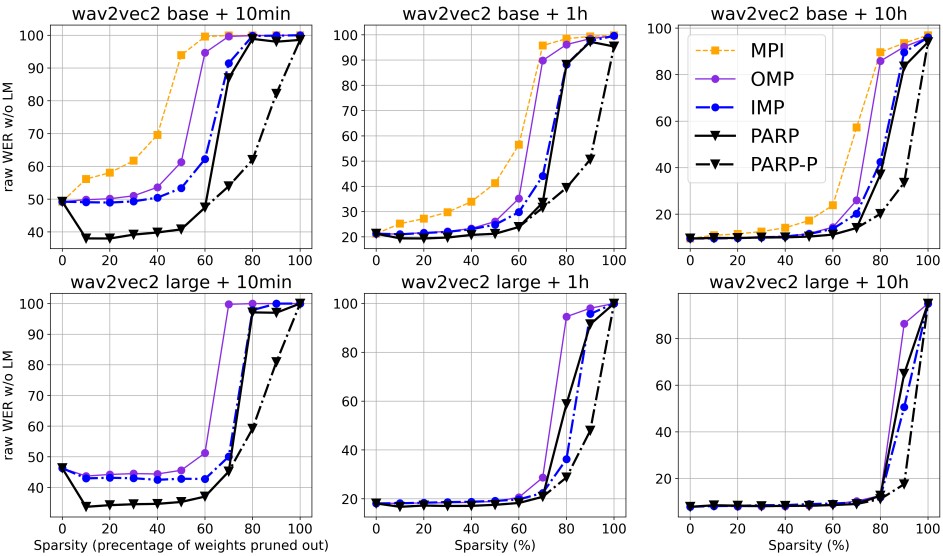

Figure 3: Comparison of different pruning techniques on LSR (`wav2vec2` with 10min/1h/10h Librispeech finetuning splits). `PARP` (black line) and `PARP-P` (black dashed line) are especially effective under ultra-low data regime (e.g. 10min) and high-sparsity (70-100%) regions.

Table 1: WER comparison of pruning LSR: `wav2vec2-base` at 90% sparsity with 10h finetuning on Librispeech without LM decoding. At 90% sparsity, `OMP`/`IMP`/`MPI` perform nearly as bad as `RP`. sub-finetuning stands for subnetwork finetuning.

| Method | # ASR finetunings | test clean | test other |
|---|---|---|---|
| RP + sub-finetuning | 1 | 94.5 | 96.4 |
| MPI + sub-finetuning | 1 | 93.6 | 96.1 |
| OMP + sub-finetuning | 2 | 92.0 | 95.3 |
| IMP + sub-finetuning | 10 | 89.6 | 93.9 |
| PARP ($90\% \rightarrow 90\%$) | 1 | 83.6 | 90.7 |
| PARP-P | | | |
| $\quad 70\% \rightarrow 90\%$ | 1 | 51.9 | 69.1 |
| $\quad 60\% \rightarrow 80\% \rightarrow 90\%$ | 2 | 33.6 | 53.3 |

Table 2: WER comparison of `PARP` for LSR with previous speech SSL results on Librispeech 10min. `PARP` discovers sparse subnetworks within `wav2vec2` with lower WER while adding minimal computational cost to the original ASR finetuning.

| Method | test clean | test other |
|---|---|---|
| Continuous BERT [3] + LM | 49.5 | 66.3 |
| Discrete BERT [3] + LM | 16.3 | 25.2 |
| `wav2vec2-base` reported [6] | 46.9 | 50.9 |
| `wav2vec2-large` reported [6] | 43.5 | 45.3 |
| `wav2vec2-base` replicated | 49.3 | 53.2 |
| `wav2vec2-large` replicated | 46.3 | 48.1 |
| `wav2vec2-base` w/ 10% PARP | 38.0 | 44.3 |
| `wav2vec2-large` w/ 10% PARP | 33.7 | 37.2 |

Next, we examine if the pruning results of LSR transfers to H2L and CSR. Figure 4 is pruning H2L and CSR with 1h of Dutch (*nl*) finetuning, and the same conclusion can be extended to other spoken languages. Comparing Figures 3 and 4, we notice that shapes of their pruning curves are different, which can be attributed to the effect of character versus phone predictions. Comparing left and center of Figure 4, we show that `PARP` and `OMP` reach 50% sparsity on H2L and 70% sparsity on CSR with minimal degradations. Furthermore, while `PARP` is more effective than `OMP` on H2L for all sparsities, such advantage is only visible in the higher sparsity regions on CSR. Lastly, Table 3 compares the subnetworks from H2L and CSR with prior work. Even with as high as 90% sparsities in either settings, subnetworks from `PARP` and `OMP` out-performs prior art.

---

[5]We underscore again that LM decoding/self-training are not included to isolate the effect of pruning.

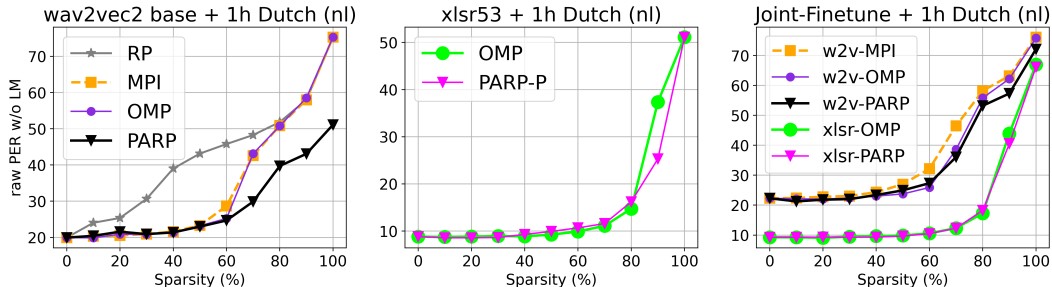

Figure 4: Comparison of pruning techniques on H2L & CSR with 1h of Dutch (*nl*) ASR finetuning. **(Left)** Pruning H2L (`wav2vec2-base` + *nl*). **(Center)** Pruning CSR (`xlsr` + *nl*). **(Right)** Pruning jointly-finetuned `wav2vec2-base` and `xlsr` on *nl*. Trend is consistent for other 9 spoken languages.

Table 3: Comparing subnetworks discovered by `OMP` and `PARP` from `wav2vec2-base` and `xlsr` with prior work on H2L and CSR. PER is averaged over 10 languages.

| Method | Pre-training | Sparsity | avg. PER |
|---|---|---|---|
| Bottleneck [38] | Babel-1070h | 0% | 44.9 |
| CPC [84] | LS-100h | 0% | 50.9 |
| Modified CPC [94] | LS-360h | 0% | 44.5 |
| `wav2vec2-base` | LS-960h | 0% | 18.7 |
| `wav2vec2 + OMP` | LS-960h | 70% | 41.3 |
| `wav2vec2 + PARP` | LS-960h | 90% | 40.1 |
| `xlsr` reported [29] | 56,000h | 0% | 7.6 |
| `xlsr` replicated | 56,000h | 0% | 9.9 |
| `xlsr + OMP` | 56,000h | 90% | 33.9 |
| `xlsr + PARP-P` | 56,000h | 90% | 22.9 |

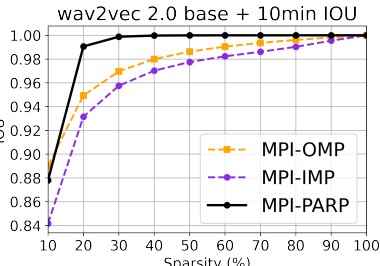

Figure 5: `PARP`'s final subnetwork and its initial `MPI` subnetwork exceeds 99.99% `IOU` after 20% sparsity (black line).

## 4.2 How Important is the Initial Subnetwork (Step 1) in `PARP`?

Obtaining a good initial subnetwork (Step 1) is critical for `PARP`, as Adjust & Re-Prune (Step 2) is operated on top of it. In this section, we isolate the effect of Step 1 from Step 2 and examine the role of the initial subnetwork in `PARP`. Figure 6 shows `PARP` with a random subnetwork from `RP`, instead of subnetwork from `MPI`, as the initial subnetwork. `PARP` with random initial subnetwork performs nearly as bad as `RP` (grey line), signifying the importance of the initial subnetwork.

Secondly, despite Observation 1, `MPI` in high sparsity regions (e.g. 90% in LSR) is not a good initial subnetwork, since the majority of the weights are already pruned out (thus is hard to be recovered from). From Figure 3, `PARP` performs only on par or even worse than `IMP` in high sparsity regions. In contrast, `PARP-P` starts with a relatively lower sparsity (e.g. 60% or 70% `MPI`), and progressively prunes up to the target sparsity. Doing so yields considerable performance gain (up to over 50% absolute WER reduction). Third, as shown in Figure 5, there is >99.99% `IOU` between the final "adjusted" subnetwork from `PARP` and its initial `MPI` subnetwork after 20% sparsity, confirming Step 2 indeed only made minimal "adjustment" to the initial subnetwork.

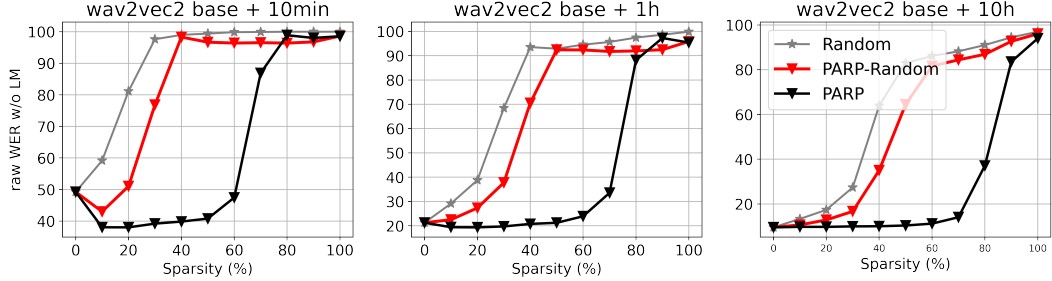

Figure 6: `PARP` with random (red line) v.s. with `MPI` (black line) initial subnetworks in LSR.

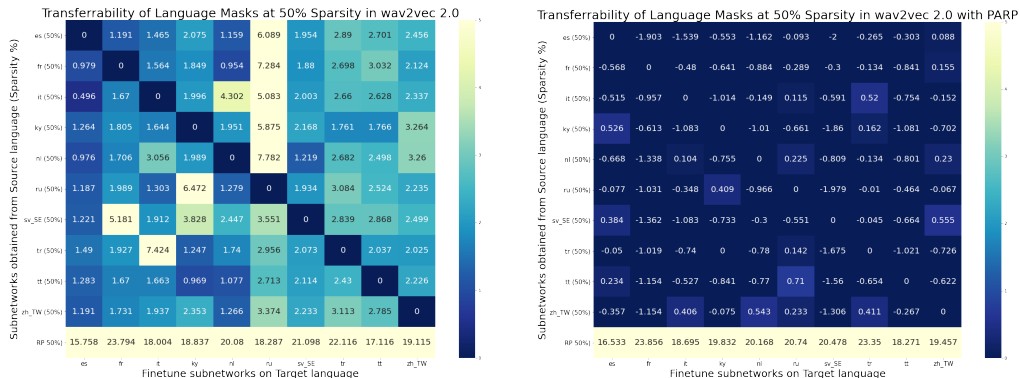

Figure 7: (**Left**) Cross-lingual `OMP` mask transfer with regular subnetwork finetuning. (**Right**) Cross-lingual `OMP` mask transfer with PARP. Last rows are RP. Values are relative PER gains over same-language pair transfer (hence the darker the bettter). Both are on H2L with pretrained `wav2vec2`. The same observation is observed on CSR with pretrained `xlsr` in Appendix 12.

### 4.3 Are Pruning Masks Transferrable across Spoken Languages?

Is it possible to discover subnetworks with the wrong guidance, and how transferrable are such subnetworks? More concretely, we investigate the transferability of `OMP` pruning mask discovered from a source language by finetuning its subnetwork on another target language. Such study should shed some insights on the underlying influence of spoken language structure on network pruning – that similar language pairs should be transferrable. From a practical perspective, consider pruning for an unseen new language in H2L, we could deploy the readily available discovered subnetworks and thus save the additional finetuning and memory costs.

In this case, the initial subnetwork of `PARP` is given by applying `OMP` on another spoken language. According to Observation 1, `PARP`'s Step 2 is effectively under-going cross-lingual subnetwork adaptation for the target language. Figure 7 shows the transferability results on H2L with pre-trained `wav2vec2-base`. On the left is a subnetwork at 50% sparsity transfer with regular finetuning that contains subtle language clusters – for example, when finetuning on *ru*, source masks from *es, fr, it, ky, nl* induces a much higher PER compare to that from *sv-SE, tr, tt, zh-TW*. On the right of Figure 7, we show that there is no cross-lingual PER degradation with `PARP`, supporting our claim above.

### 4.4 Discovering a Single Subnetwork for 10 Spoken Languages

A major downside of pruning pre-trained SSL models for many downstream tasks is the exponential computational and memory costs. In H2L and CSR, the same pruning method needs to be repeatedly re-run for each downstream spoken language at each given sparsity. Therefore, we investigate the possibility of obtaining a single shared subnetwork for all downstream languages. Instead of finetuning separately for each language, we construct a joint phoneme dictionary and finetune `wav2vec2` and `xlsr` on all 10 languages jointly in H2L and CSR. Note that `PARP` with joint-finetuning can retrieve a shared subnetwork in a single run. The shared subnetwork can then be decoded for each language separately. The right side of Figure 4 illustrates the results.

Comparing joint-finetuning and individual-finetuning, in H2L, we found that the shared subnetwork obtained via `OMP` has lower PERs between 60∼80% but slightly higher PERs in other sparsity regions; in CSR, the shared subnetwork from `OMP` has slightly worse PERs at all sparsities. Comparing `PARP` to `OMP` in joint-finetuning, we found that while `PARP` is effective in the individual-finetuning setting (left of Figure 4), its shared subnetworks are only slightly better than `OMP` in both H2L and CSR (right of Figure 4). The smaller performance gain of `PARP` over `OMP` in pruning jointly-finetuned models is expected, since the important weights for each language are disjoint and joint-finetuning may send mixed signal to the adjustment step in `PARP` (see Figure 8 for better illustration).

### 4.5 Does `PARP` work on Pre-trained BERT/XLNet?

We also analyzed whether Observation 1 holds for pre-trained BERT/XLNet on 9 GLUE tasks. Surprisingly, we found that there are also high (>98%) overlaps between the 9 tasks' `IMP` pruning masks. Given this observation, we replicated the cross-task subnetwork transfer experiment (take subnetwork found by `IMP` at task A and finetune it for task B) in BERT-Ticket [19] on pre-trained BERT/XLNet with `PARP`. Table 4 compares `PARP` (averaged for each target task) to regular finetuning,

hinting the applicability of `PARP` to more pre-trained NLP models and downstream natural language tasks. Detailed scores and figures are in Appendix 13.

Table 4: Comparison of cross-task transfer on GLUE (subnetwork from source task A is finetuned for target task B). Numbers are averaged acc. across source tasks for each target task.

| Method | Averaged transferred subnetworks performance finetuned for | | | | | | | | |
|---|---|---|---|---|---|---|---|---|---|
| | CoLA | MRPC | QNLI | QQP | RTE | SST-2 | STS-B | WNLI | MNLI |
| 70% sparse subnetworks from pre-trained BERT | | | | | | | | | |
| Same-task Transfer (top line) | 38.89 | 75.57 | 88.89 | 89.95 | 58.37 | 89.99 | 87.34 | 53.87 | 82.56 |
| Cross-task Transfer with `PARP` | **28.48** | **75.98** | **87.12** | **90.40** | **59.69** | **89.59** | **86.25** | **54.62** | **81.61** |
| Regular Cross-task Transfer [19] | 10.12 | 71.94 | 86.54 | 88.50 | 57.59 | 88.80 | 80.27 | 54.03 | 80.48 |
| 70% sparse subnetworks from pre-trained XLNet | | | | | | | | | |
| Same-task Transfer (top line) | 29.92 | 76.47 | 89.62 | 90.74 | 59.21 | 92.2 | 80.78 | 42.25 | 85.16 |
| Cross-task Transfer with `PARP` | **30.09** | **77.56** | **87.10** | **90.66** | **58.88** | **91.73** | **83.80** | **52.11** | **83.87** |
| Regular Cross-task Transfer [19] | 11.47 | 74.16 | 85.21 | 89.11 | 55.80 | 90.19 | 75.61 | 42.25 | 82.65 |

Figure 8: Conceptual sketch of pruning the few task-specific important weights in pretrained SSL. **(A)** Task-aware subnetwork discovery(`OMP/IMP`) is more effective than task-agnostic pruning (`MPI`) since it foresees the important weights in advance, via multiple downstream finetunings. **(B)** `PARP` starts with an initial subnetwork given by `MPI`. Observation 1 suggests that the subnetwork is only off by the few important weights, and thus Step 2 revives them by adjusting the initial subnetwork.

### 4.6 Implications

Observation 1 is consistent with the findings of probing large pre-trained NLP models, that pre-trained SSL models are over-parametrized and there exist task-oriented weights/neurons. Figure 2 implies that these important weights only account for a small part of the pre-trained speech SSL. In fact, a large body of NLP work is dedicated to studying task-oriented weights in pre-trained models. To name a few, [37, 35, 7, 115] measured, [7, 34, 61] leveraged, [81, 46] visualized, and [105, 36, 13] pruned out these important weights/neurons via probing and quantifying contextualized representations. Based on Observation 1, we can project that these NLP results should in general transfer to speech, see pioneering studies [9, 8, 24, 23]. However, different from them, `PARP` leverages important weights for `UMP` on the whole network structure instead of just the contextualized representations.

We could further hypothesize that a good pruning algorithm avoids pruning out task-specific neurons in pre-trained SSL [67, 48, 79], see Figure 8. This hypothesis not only offers an explanation on why `PARP` is effective in high sparsity regions and cross-lingual mask transfer, it also suggests that an iterative method such as `IMP` is superior to `OMP` because `IMP` gradually avoids pruning out important weights in several iterations, at the cost of more compute[6]. Finally, we make connections to prior work that showed `RP` prevail [11, 19, 75, 77, 92] – under a certain threshold and setting, task-specific neurons are less likely to get "accidentally" pruned and thus accuracy is preserved even with `RP`.

---

[6]From Section 6 of [39]: "iterative pruning is computationally intensive, requiring training a network 15 or more times consecutively for multiple trials." From Section 1 of [48]: "several iterations of alternate pruning and retraining are necessary to get a fair compression rate on AlexNet, while each retraining process consists of millions of iterations, which can be very time consuming."

## 5  Related Work

**Modern Speech Paradigm and ASR Pruning.** As model scale [101, 6, 50, 47, 124, 90, 89, 125, 16, 121, 68] and model pre-training [6, 127, 29, 60, 57, 63, 55, 118, 14, 58, 96, 95, 83, 86, 109] have become the two essential ingredients for obtaining SOTA performance in ASR and other speech tasks, applying and developing various forms of memory-efficient algorithms, such as network pruning, to these large-scale pre-trained models will predictably soon become an indispensable research endeavor. Early work on ASR pruning can be dated back to pruning decoding search spaces [1, 91, 100, 52, 116, 128] and HMM state space [103]. Since the seminal work of Yu et al. [122], ASR pruning has focused primarily on end-to-end network architecture: [98, 114] applied pruning and quantization to LSTM-based RNN-Transducers, [85] applied knowledge distillation to Conformer-based RNN-Transducers, [104, 99, 70] designed efficient architecture/mechanisms for LSTM, Transformer, Conformer-based ASR models, [82] applied pruning to Deep Speech, [12] introduced SNR-based probabilistic pruning on LSTM-based CTC model, [43] proposed entropy-regularizer for LSTM-based ASR model, [119, 87] applied SVD on ASR models' weight matrices. We emphasize that our work is the first on pruning large self-supervised pre-trained models for low-resource and multi-lingual ASR. In addition, to our knowledge, none of the prior speech pruning work demonstrated the pruned models attain superior performance than its original counterpart.

## 6  Conclusions

We introduce PARP, a simple and intuitive pruning method for self-supervised speech recognition. We conduct extensive experiments on pruning pre-trained wav2vec 2.0 and XLSR-53 under three low-resource settings, demonstrating (1) PARP discovers better subnetworks than baseline pruning methods while requiring a fraction of their computational cost, (2) the discovered subnetworks yields over 10% WER reduction over the full model, (3) PARP induces minimal cross-lingual subnetwork adaptation errors, (4) PARP can discover a shared subnetwork for multiple spoken languages in one pass, and (5) PARP significantly reduces cross-task adaptation errors of pre-trained BERT/XLNet. Beyond the scope of our study, we aspire PARP as the beginning of many future endeavours on developing more efficient speech SSL models.

**Broader Impact.** The broader impact of this research work is making speech technologies more accessible in two orthogonal dimensions: (i) extending modern-day speech technology to many under-explored low-resource spoken languages, and (ii) introducing a new and flexible pruning technique to current and future speech SSL frameworks that reduces the computational costs required for adapting (finetuning) them to custom settings. We do not see its potential societal harm.

## Limitations and Future Work

We make clear of the major limitations of our work, and the full list is in Appendix 19. The basis of all the pruning methods in the study is unstructured magnitude weight pruning. Although sparsity is explicitly enforced in the models, we do not suggest that the sparse models are more memory or energy efficient than the original dense models. We do believe that our methodology and results should provide meaningful insights and be easily extended upon to more advanced unstructured or structured pruning methods. We are also curious of the possibility of finetuning or storing modern speech SSL models on local hardware devices.

Results on cross-lingual mask transfer on pre-trained wav2vec 2.0 in Section 4.3 is limited to ASR. We do not claim pruning masks to be transferrable across speech tasks (e.g. prune wav2vec2 for speaker ID and transfer for ASR). We provide a pilot cross-task mask transfer study on 3 speech tasks (phone recognition, speaker recognition, slot-filling) in SUPERB [120], and results is in Appendix 16.

We claim PARP *could* improve the downstream ASR performance over the full wav2vec 2.0, yet we do not claim it as a plug-and-play method into any SOTA ASR pipeline, such as [126], to get a performance boost. We provide a preliminary experiment on combining PARP and transformer-LM decoding in Appendix 15. Nonetheless, due to resource limitations and to isolate the effect of pruning, it remains upon investigations on the complete effects of speech pruning in different setups.

## Acknowledgments

We thank IBM for the donation to MIT of the Satori GPU cluster, and John Cohn for maintaining the cluster. We also thank Lucy Chai, Wei-Ning Hsu, Desh Raj, Shu-wen Leo Yang, Abdelrahman Mohamedm, Erica Cooper, and anonymous reviewers for helpful suggestions and paper editing. This work is part of the low-resource language learning project funded by the MIT-IBM Waston AI Lab.

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
