# Supplementary Material

## Contents


# 8 Model Details

Model and pruning configurations for `wav2vec2-base`, `wav2vec2-large`, and `xlsr` can be found in Section 8.1. Fintuning hyper-parameters are generally the same as in [6], and we detailed them in Section 8.2. `PARP`'s hyper-parameter is detailed in Section 8.3. More details on system implementations is in Section 8.4.

## 8.1 Model and Pruning Configurations

wav2vec 2.0 consists of three modules: a 7-layer CNN feature encoder for pre-processing raw speech waveforms, a quantization layer for discretizing, and a BERT for learning contextualized representations. Given that the feature encoder is fixed and the quantization layer is discarded during finetuning, we focus on pruning the BERT module in wav2vec 2.0 and XLSR-53. We also do not prune the positional embedding layer nor the layer normalization layers within BERT. This setup is consistent with BERT-Ticket [19]. wav2vec 2.0 BASE (`wav2vec2-base`) is based on BERT-BASE, which has 12 transformer blocks, hidden dimension 768, 12 self-attention heads, and 95M parameters. wav2vec 2.0 LARGE (denote as `wav2vec2-large`) is based on BERT-LARGE, which has 24 transformer blocks, hidden dimension 768, 16 self-attention heads, and 315M parameters. XLSR-53 (denoted as `xlsr`) shares the same architecture as `wav2vec2-large`. We took `wav2vec2-base` and `wav2vec2-large` that were pre-trained on Librispeech 960h. `wav2vec2-base`, `wav2vec2-large`, and `xlsr` are pre-trained with the contrastive predictive coding objective.

**More on Pruning Configuration.** There are 3 components in `wav2vec2/xlsr` that we did not prune out: (1) CNN feature extractor, (2) layer norm running statistics, and (3) positional embedding/task-specific linear layer. For (1), it is due to the CNN feature extractor being fixed during finetuning by default, and the majority of the model parameters lie in the BERT module in `wav2vec2/xlsr`. For (2)(3), we simply follow the setup described in BERT-Ticket [19]. These 3 decisions is why in left of Figure 4, `PARP` (black line) attains $\sim$50% PER at 100% sparsity. In fact, while re-producing BERT-Ticket [19], we were surprised that BERT's layer norm statistics plus its final linear layer achieve non trivial loss/accuracy (e.g. BERT's MLM at 0% sparsity is $\sim$60% accuracy while at 100% sparsity is $\sim$15% accuracy.).

## 8.2 Finetuning Hyper-Parameters

`wav2vec2` is finetuned for 20k steps on the 10h split, 15k steps on the 1h split, and 12k steps on the 10min split. `xlsr` is finetuned for 12k steps for each spoken languages. In the default setup in [6], `wav2vec2` except the final linear layer is freezed for 10k steps, however, we observe doing so on the pruned models may lead to training instability. Therefore, we do not include this trick in our fine-tuning setups. The learning rate ramps up linearly for first 10% of the steps, remains the same for 40% of the steps, and decay exponentially for 50% of the steps. The waveform encoder output is randomly masked according to [6]. For LSR, the validation set is the dev-other subset from Librispeech.

## 8.3 `PARP` Hyper-Parameters

PARP introduces an additional pruning frequency hyper-parameter, $n$ in Algorithm Table 1. As long as $n$ is a sensible small number (e.g. 5-50 out of 10k+ steps), the final pruned models should have similar performance. We heuristically set $n = 5$ for pruning XLSR on all spoken language splits; we set $n = 50$ for `wav2vec2-base` on 10min/1h, $n = 5$ for `wav2vec2-base` on 10h, $n = 5$ for `wav2vec2-large` on 10min, $n = 2$ for `wav2vec2-large` on 1h, and $n = 1$ for `wav2vec2-large` on 10h.

## 8.4 Implementation

All experiments are based on the Fairseq repository[7] and Wav2letter++ decoding[8]. We took publicly available pre-trained `wav2vec2-base`, `wav2vec2-large`, and `xlsr`[9]. The pruning code is based on

---

[7] https://github.com/pytorch/fairseq
[8] https://github.com/flashlight/wav2letter
[9] Pre-trained models available at https://github.com/pytorch/fairseq/blob/master/examples/wav2vec/README.md

PyTorch's pruning module [10]. For each experiment, we fine-tune the model on either 2 or 4 GPUs in parallel, and unlike the standard wav2vec 2.0 fine-tuning setup, we do not include a LM for validation during fine-tuning. Given that not all of our GPUs support FP16, our fine-tuning setup is on FP32. For fair comparison, we imposed a reasonable computational budget for all pruning methods used in this study[11].

## 9    Experimental Setup for LSR, H2L, and CSR

For LSR, we finetune pre-trained `wav2vec2-base` and `wav2vec2-large` on the 10h/1h/10min splits from Librispeech and Libri-light, as this is the *de facto* setup for studying speech representation learning [6]. For H2L, we replicate the setting described in [94, 29], where pre-trained `wav2vec2-base` is finetuned on 10 spoken languages (1 hour each) from CommonVoice: *Spanish (es), French (fr), Italian (it), Kyrgyz (ky), Dutch (nl), Russian (ru), Swedish (sv-SE), Turkish(tr), Tatar (tt), and Mandarin (zh-TW)*. For CSR, we replicate the setting in [29], where pre-trained `xlsr` is finetuned on the same 10 languages as in H2L. Studying LSR can inform us the effect of amount of finetuning supervision (10min∼10h) and pre-trained model scales (`base` v.s. `large`) on pruning; on the other hand, comparing CSR and H2L could yield insights on the effect of mono-lingual versus cross-lingual pre-training on pruning.

**Evaluation Criteria.** Word Error Rate (WER) is reported for LSR; Phone Error Rate (PER) is reported for H2L and CSR[12]. Earlier work on pruning sequence to sequence tasks, such as ASR [12] or Machine Translation [123, 41], showed that pruned models do not match or outperform the full model, albeit with "minimal degradation". Moreover, to isolate the effects of different pruning methods, we **do not** include any external LM nor any means of self-training [118] during training or decoding. To provide an unbiased grounding and accurate reflection of the pruned models, we thus report relative gains of our proposed method over `OMP/IMP/MPI`, in addition to their raw WER/PERs.

## 10    How important is the `IMP` rewinding starting point?

We also examined the effectiveness of `IMP` rewinding [40, 93] for pruning speech SSL, where instead of re-starting each `IMP` pruning iteration all the way back from pre-trained SSL initializations, the iteration starts at some points during the downstream ASR finetuning. For example, in figure 9, `IMP` with 10% rewinding (dark red line) means that each pruning iteration starts at 10% into the ASR downstream finetuning; We find that rewinding has minimal effect for pruning speech SSL, which aligns with the results in NLP [19]. Curiously, we observe the effect diminishes when the pre-training model size is scaled up from `base` to `large`.

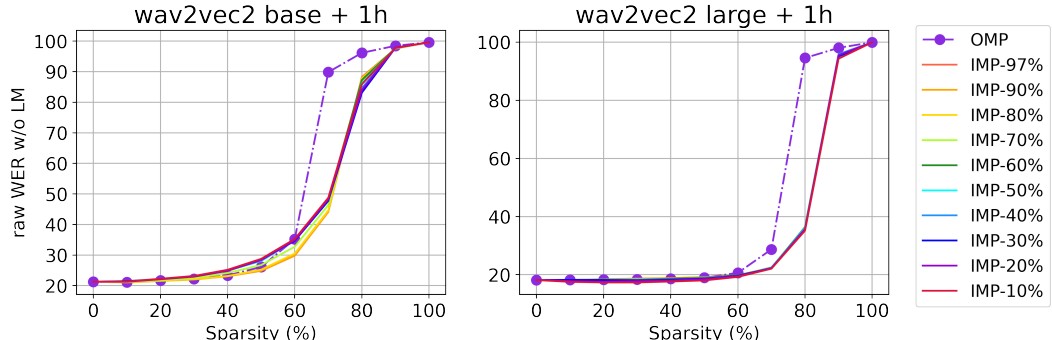

Figure 9: `IMP` on `wav2vec2-base` and `wav2vec2-large` with different rewinding starting point within the downstream ASR finetuning. Its effect diminishes when pruning `wav2vec2-large`.

---

[10]`https://pytorch.org/tutorials/intermediate/pruning_tutorial.html`

[11]Each finetuning run is capped at a total of 100 V100 hours. For example, `OMP` requires 2 finetunings, so we will run it for at most a total of 50 hours on across 4 V100s.

[12]WER/PER (lower the better) is standard criteria for ASR. This is opposite to previous work on pruning CV or NLP models, where accuracy or BLEU scores (higher the better) was reported.

# 11 `OMP` Masks Overlap in H2L and CSR

We provide the rest of Figure 2 at other sparsities to support Observation 1. For readability, we re-state it again:

> *For any sparsity, any amount of finetuning supervision, any pre-training model scale, and any downstream spoken languages, the non-zero ASR pruning masks obtained from task-agnostic subnetwork discovery has high `IOU`s with those obtained from task-aware subnetwork discovery.*

In addition to `IOU`, we also provide the overlap percentage between masks[13]. We divide this section into `OMP` masks overlap over spoken language pairs on finetuned `wav2vec2-base` in H2L (Section 11.1) and overlaps on finetuned `xlsr` in CSR (Section 11.2).

## 11.1 `OMP` Masks Overlap in H2L

**H2L `OMP` masks overlap procedure.** Each set of experiments require 10×10 rounds of `xlsr` finetunings because there are 10 downstream spoken languages ASR. The experimental procedure is:

1. Finetune `wav2vec2-base` for a source spoken language ASR.
2. Prune the finetuned model and obtain an `OMP` mask for each spoken language ASR.
3. Calculate `IOU`/mask overlap over all pairs of spoken language masks at each sparsity.

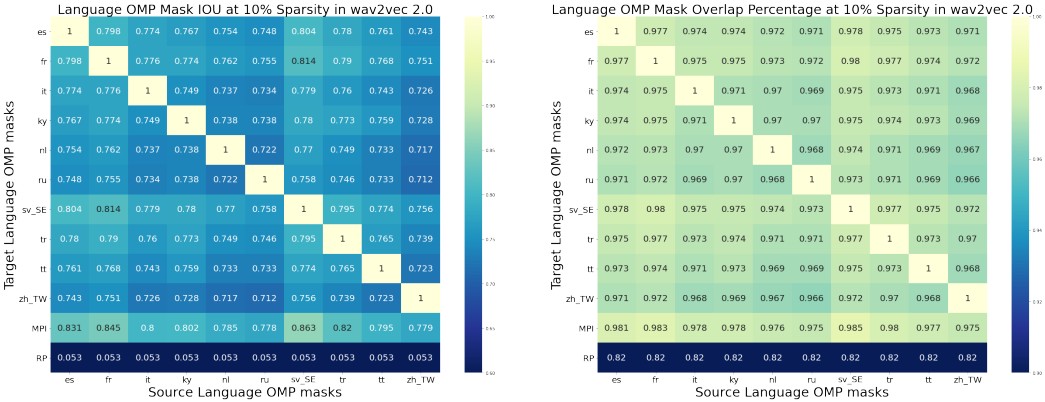

Figure 10: `OMP` pruning masks `IOU`s and overlap percentages on finetuned `wav2vec2` at 10% sparsity.

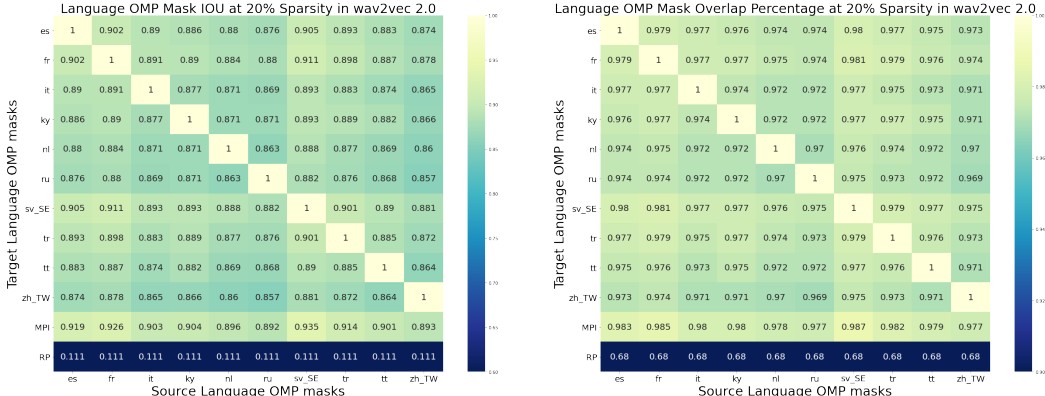

Figure 11: `OMP` pruning masks `IOU`s and overlap percentages on finetuned `wav2vec2` at 20% sparsity.

---

[13]Instead of taking the Union in the denominator as in `IOU`, simply take the full number of parameters.

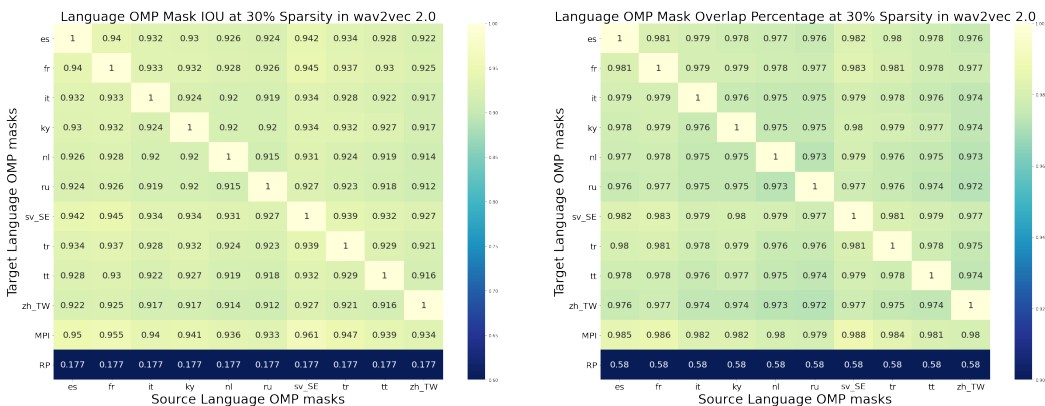

Figure 12: OMP pruning masks IOUs and overlap percentages on finetuned wav2vec2 at 30% sparsity.

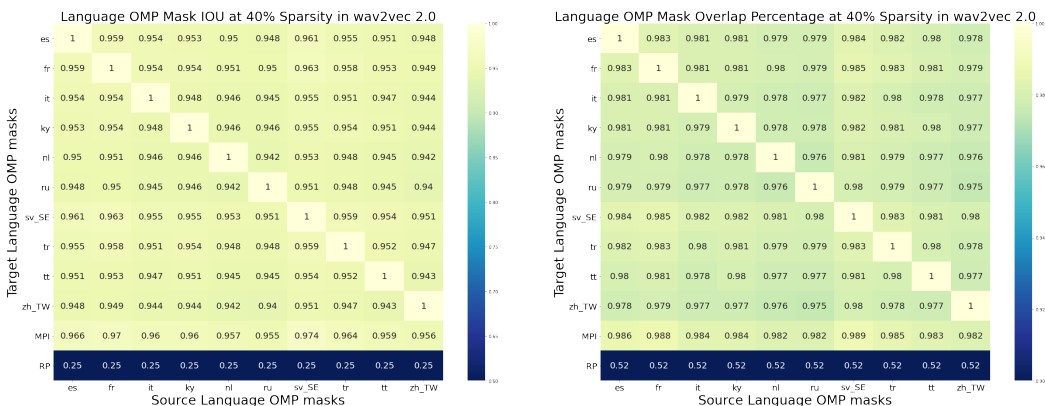

Figure 13: OMP pruning masks IOUs and overlap percentages on finetuned wav2vec2 at 40% sparsity.

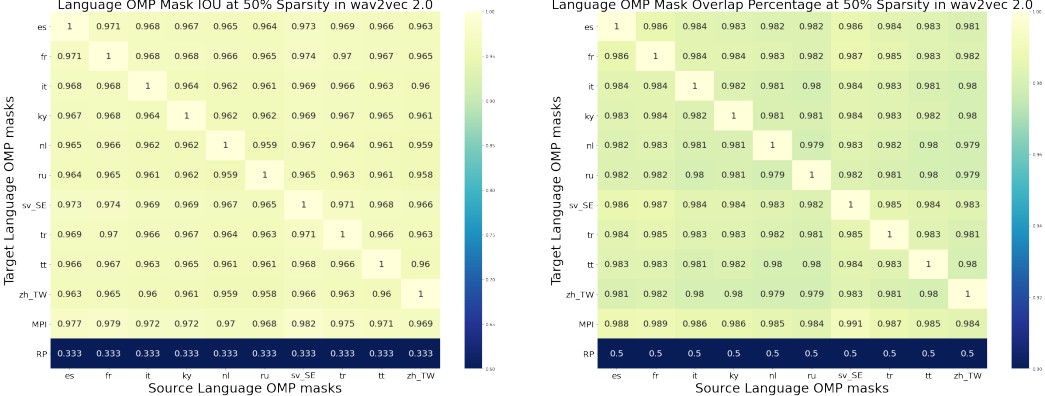

Figure 14: OMP pruning masks IOUs and overlap percentages on finetuned wav2vec2 at 50% sparsity.

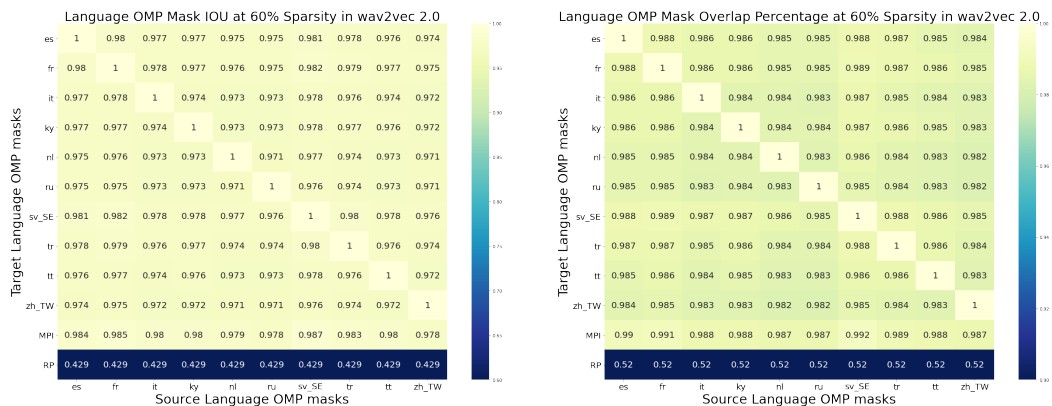

Figure 15: `OMP` pruning masks `IOU`s and overlap percentages on finetuned `wav2vec2` at 60% sparsity.

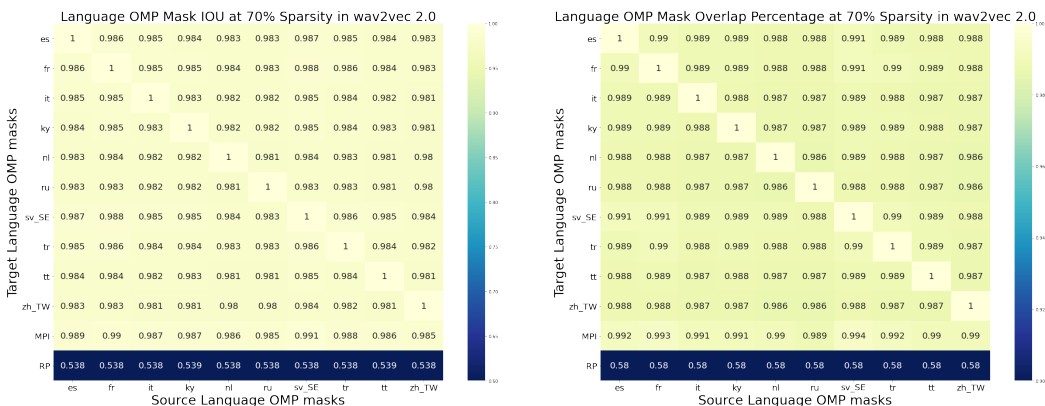

Figure 16: `OMP` pruning masks `IOU`s and overlap percentages on finetuned `wav2vec2` at 70% sparsity.

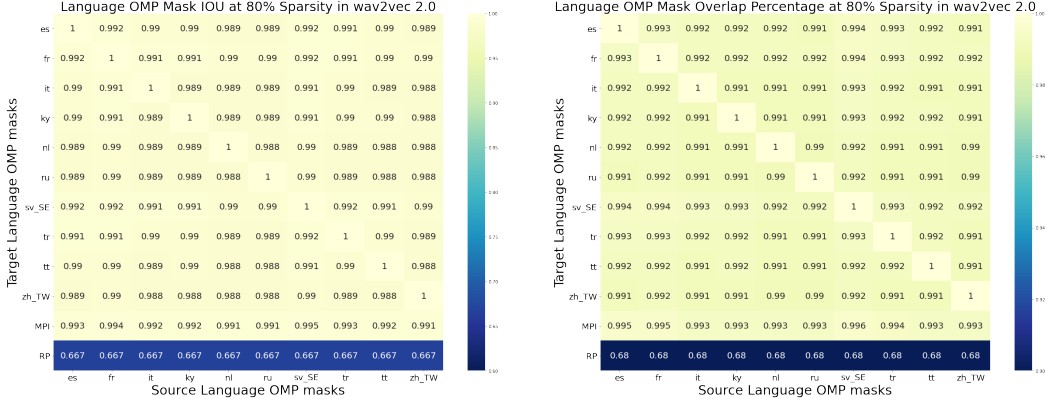

Figure 17: `OMP` pruning masks `IOU`s and overlap percentages on finetuned `wav2vec2` at 80% sparsity.

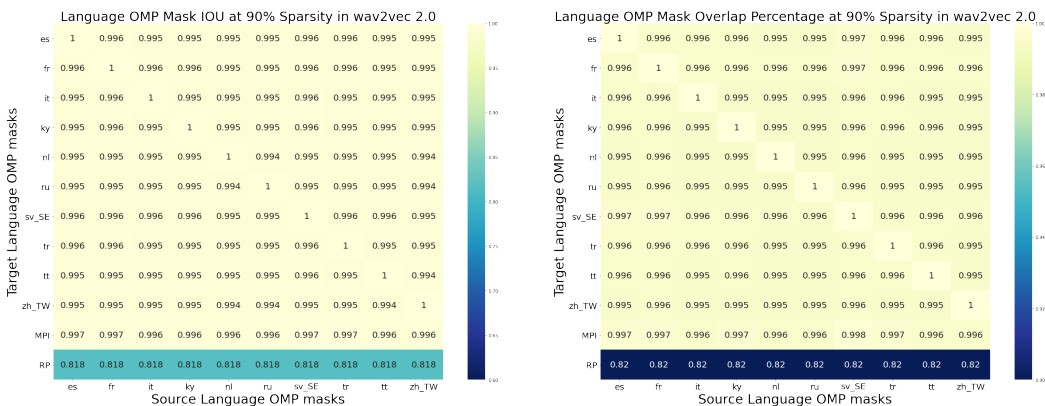

Figure 18: `OMP` pruning masks `IOUs` and overlap percentages on finetuned `wav2vec2` at 90% sparsity.

## 11.2 `OMP` Masks Overlap in CSR

**CSR `OMP` masks overlap procedure.** Each set of experiments require 10×10 rounds of `xlsr` finetunings because there are 10 downstream spoken languages ASR. The experimental procedure is:

1. Finetune `xlsr` for a source spoken language ASR.
2. Prune the finetuned model and obtain an `OMP` mask for each spoken language ASR.
3. Calculate `IOU`/mask overlap over all pairs of spoken language masks at each sparsity.

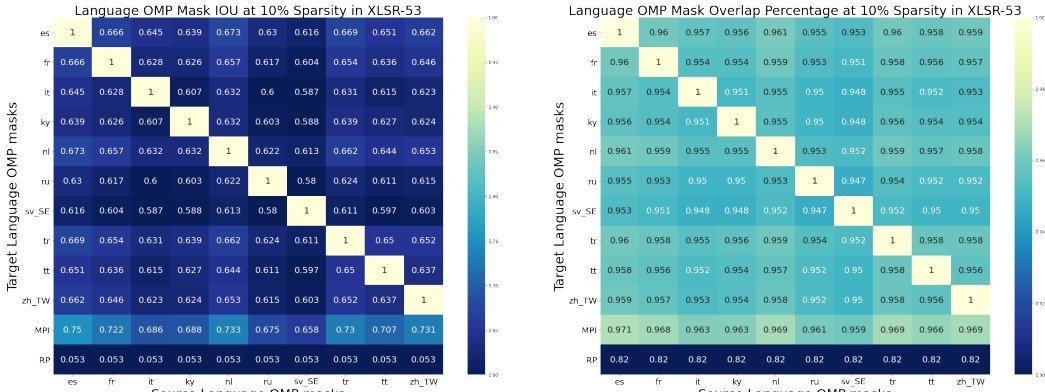

Figure 19: `OMP` pruning masks `IOU`s and overlap percentages on finetuned `xlsr` at 10% sparsity.

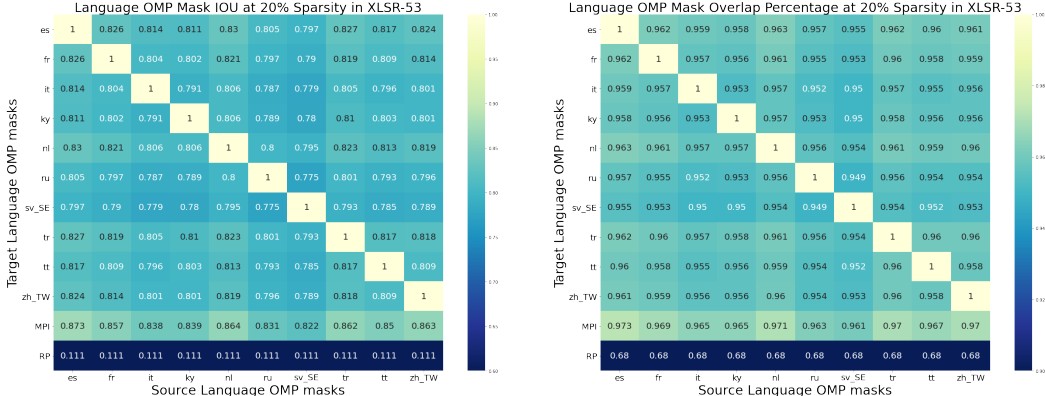

Figure 20: `OMP` pruning masks `IOU`s and overlap percentages on finetuned `xlsr` at 20% sparsity.

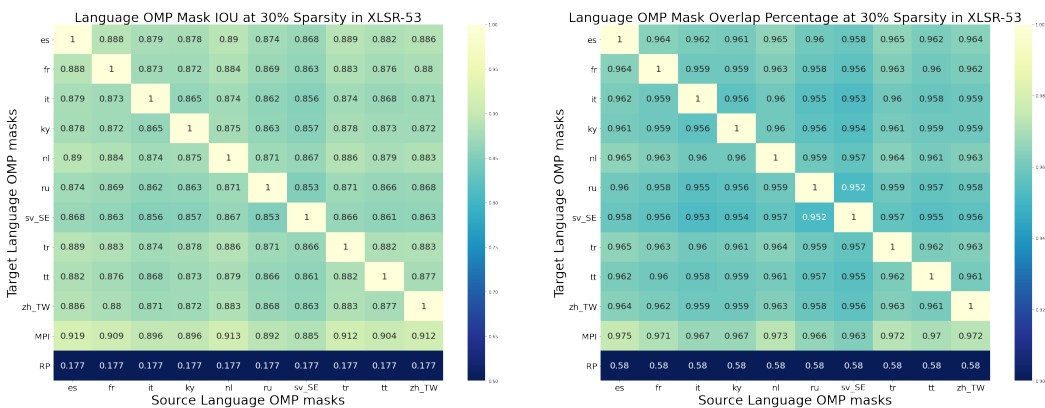

Figure 21: `OMP` pruning masks `IOUs` and overlap percentages on finetuned `xlsr` at 30% sparsity.

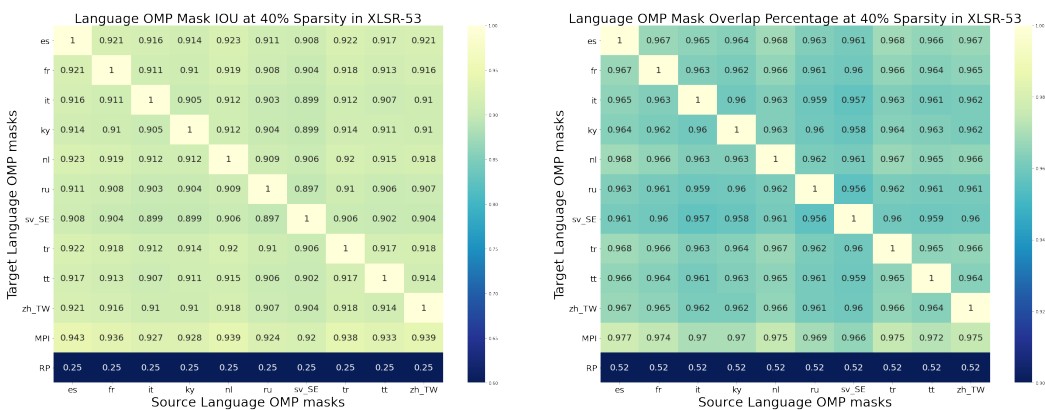

Figure 22: `OMP` pruning masks `IOUs` and overlap percentages on finetuned `xlsr` at 40% sparsity.

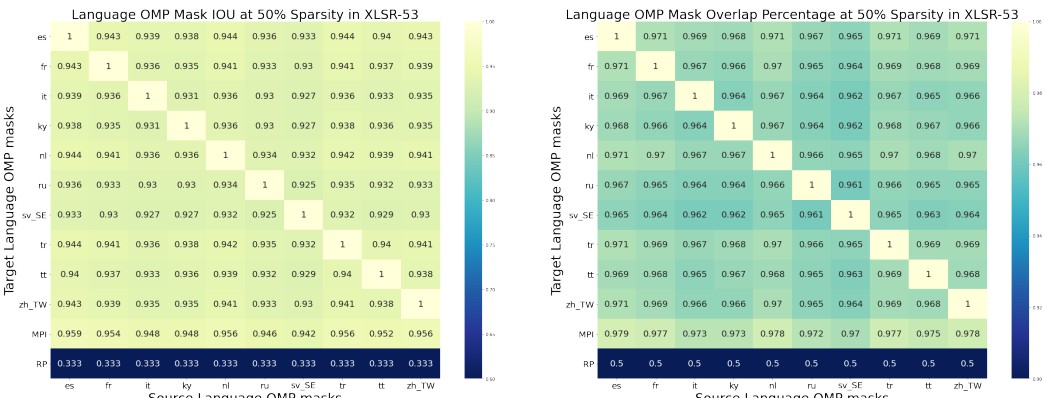

Figure 23: `OMP` pruning masks `IOUs` and overlap percentages on finetuned `xlsr` at 50% sparsity.

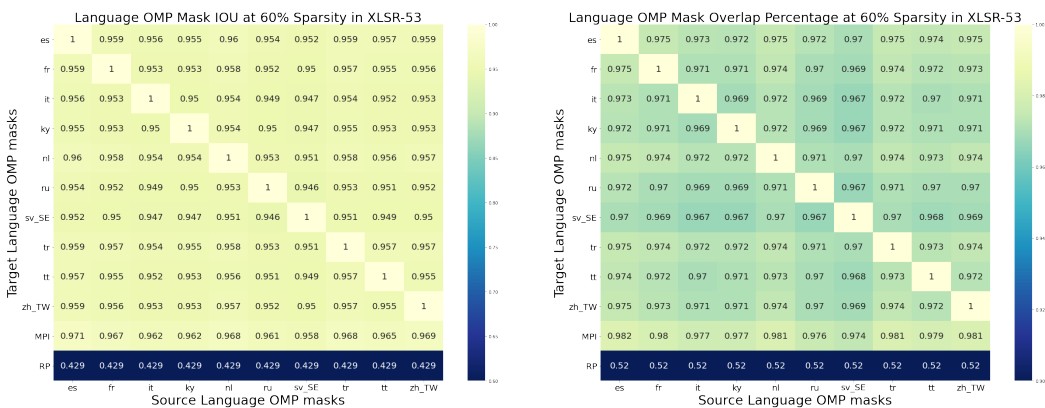

Figure 24: `OMP` pruning masks `IOUs` and overlap percentages on finetuned `xlsr` at 60% sparsity.

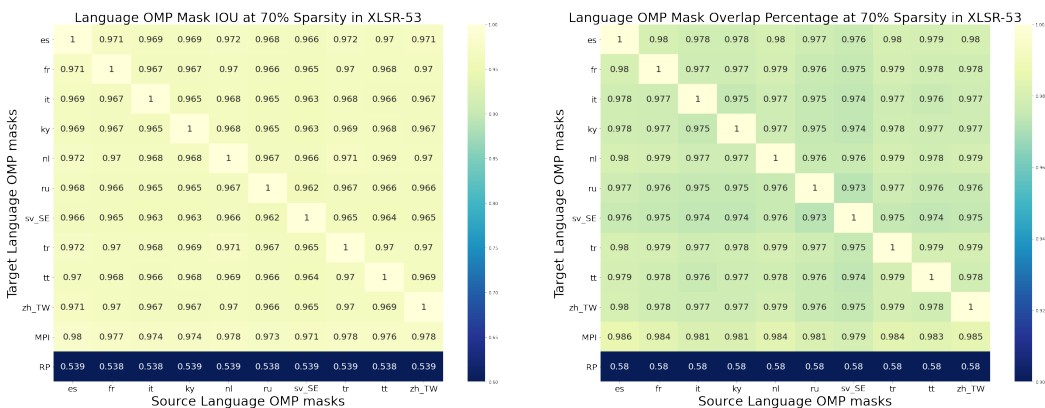

Figure 25: `OMP` pruning masks `IOUs` and overlap percentages on finetuned `xlsr` at 70% sparsity.

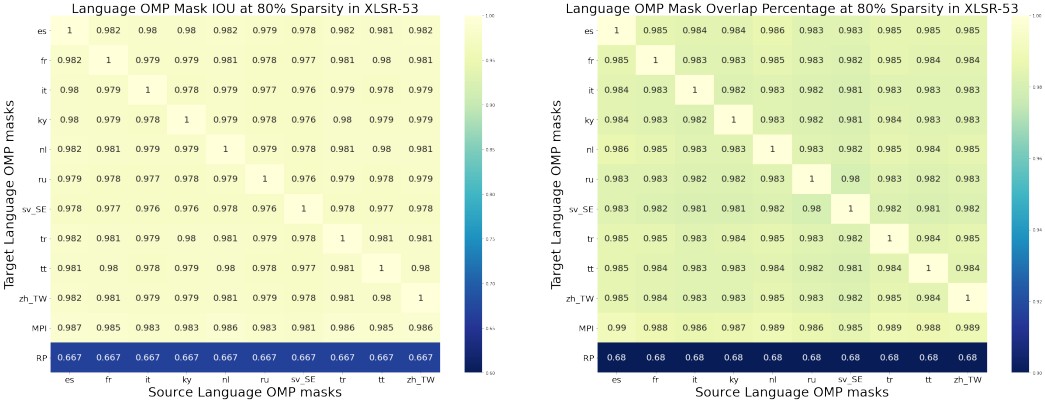

Figure 26: `OMP` pruning masks `IOUs` and overlap percentages on finetuned `xlsr` at 80% sparsity.

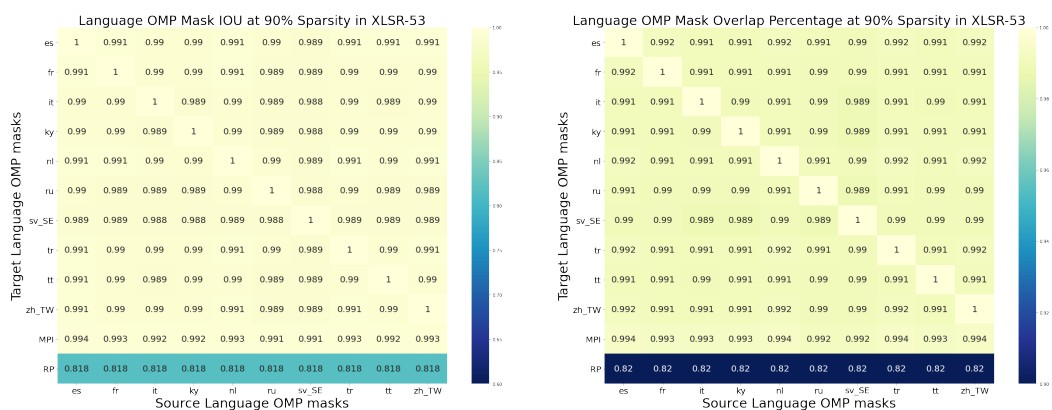

Figure 27: `OMP` pruning masks `IOUs` and overlap percentages on finetuned `xlsr` at 90% sparsity.

## 12  `xlsr` Cross-Lingual Mask Transfer

**Cross-lingual mask transfer procedure.** Each set of experiments require $10 \times 10 \times 2$ rounds of `xlsr` finetunings because there are 10 downstream spoken languages ASR, and we finetune for each spoken language ASR twice (the first one for retrieving mask, and second one for mask transfer). The experimental procedure is:

1. Finetune `xlsr/wav2vec2` for a source spoken language ASR.
2. Prune the finetuned model and obtain an `OMP` mask for each spoken language ASR.
3. Apply the `OMP` mask at `xlsr` pre-trained initializations and finetune for a target spoken language ASR with `PARP`.

Figure 28 is the result, and it has the same cross-lingual mask transfer setup as that in Section 4.3 and Figure 7, except the pre-trained model is `xlsr` instead of `wav2vec2`.

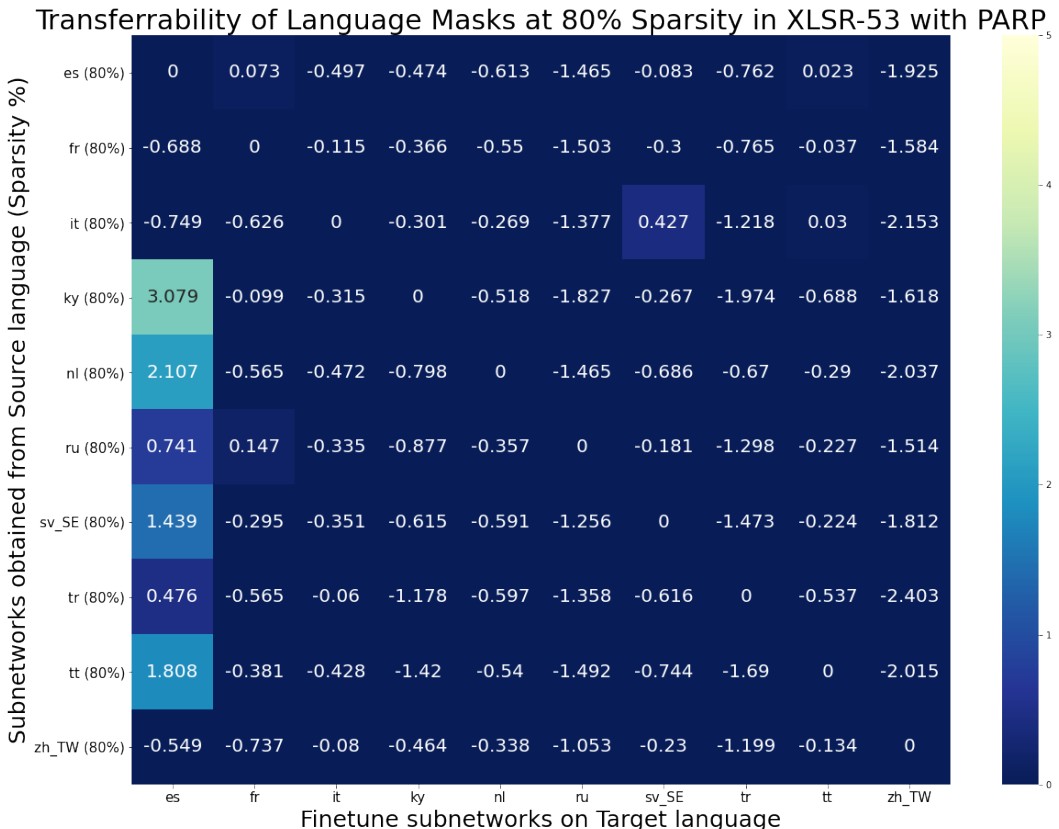

Figure 28: Cross-lingual mask transfer for `xlsr`. Cross-lingual mask transfer with `PARP` has minimal PER degradation (darker the better).

# 13 Details of Task Transfer Results on Pre-trained BERT

**Cross-task mask transfer procedure.** Each set of experiments require $9 \times 9 \times 2$ rounds of finetunings because there are 9 subtasks in GLUE, and we finetune for each subtask twice (the first one for retrieving mask, and second one for mask transfer). We first note that our cross-task transfer experimental designs are closely knitted to NLP probing work's experimental setup [113, 36], i.e. pretrained BERT/XLNet on 9 subtasks in GLUE. The experimental procedure is:

1. Finetune BERT/XLNet for a source task in GLUE.
2. Prune the finetuned model and obtain an `IMP` mask for each task.
3. Apply the `IMP` mask at BERT/XLNet pre-trained initializations and finetune for a target task in GLUE with `PARP`.

Figure 29 is the `IMP` mask overlap for pre-trained BERT on the 9 natural language tasks in GLUE. Figure 30 is the cross-task transfer result. For all the GLUE tasks, `PARP` can achieve better results compared to BERT-Ticket (cross-task subnetwork regular finetuning) [19]. For the tasks with poor transferability in BERT-Ticket [19], like CoLA and STS-B, `PARP` still achieves good transfer scores.

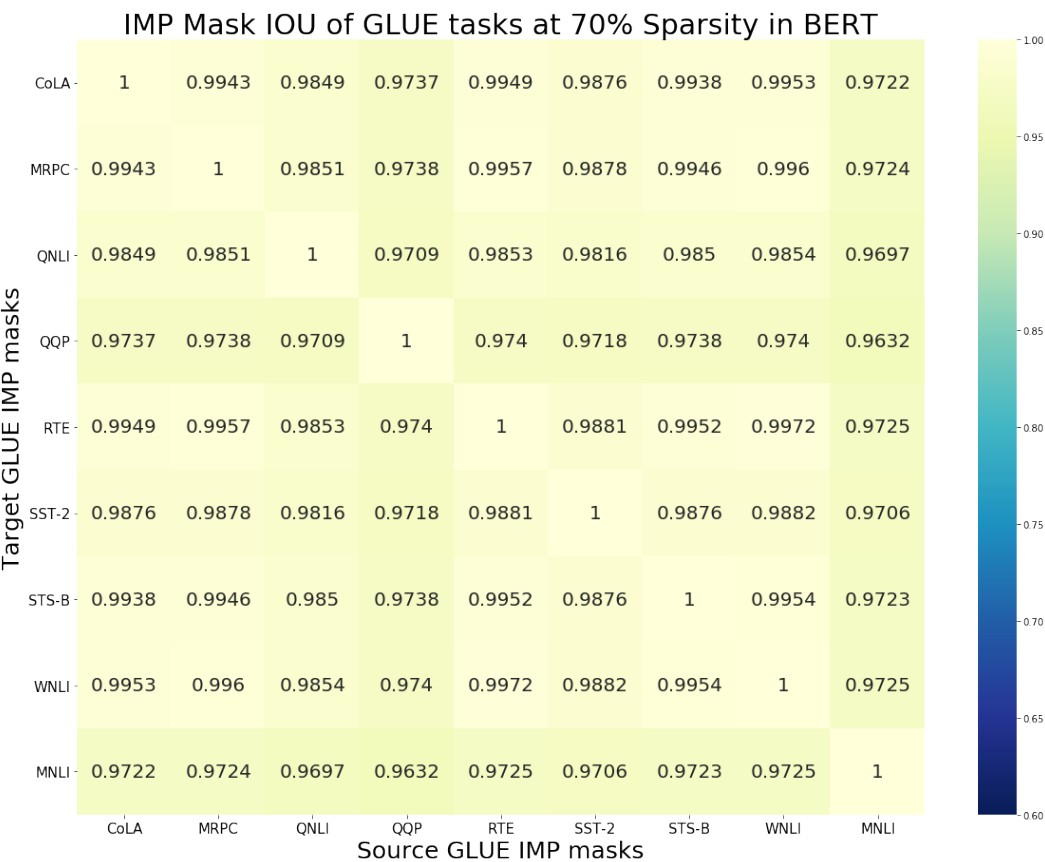

Figure 29: `IOU`s over all GLUE tasks' `IMP` pruning masks on finetuned BERT at 70% sparsity. Notice the high overlap rates, which aligns with Observation 1.

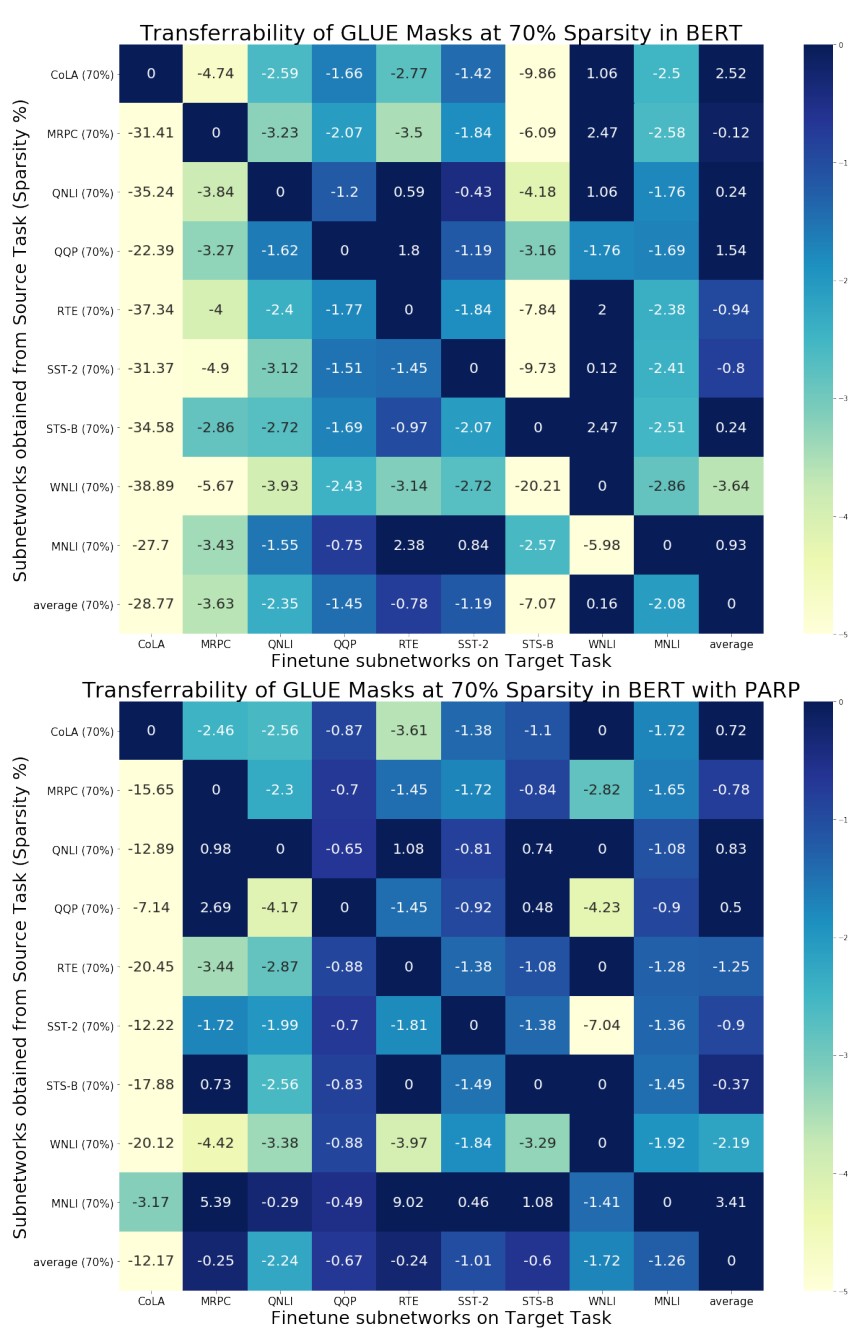

Figure 30: Results for subnetwork transfer experiment (take subnetwork found by IMP at task A and finetune it for task B). **Top:** the transfer results in BERT-Ticket [19]. **Bottom:** transfer with PARP finetuning instead. Each row is a source task A, and each column is a target task B. All numbers are subtracted by the scores of same-task transfer (task A = task B, and the darker the better).

# 14 Full H2L and CSR Pruning Results

We provide the full set of H2L and CSR pruning (refer to Section 4.1 and Section 4.4 for experimental description). Below are the rest of Figure 4 to other spoken languages from CommonVoice: *Spanish (es), French (fr), Italian (it), Kyrgyz (ky), Dutch (nl), Russian (ru), Swedish (sv-SE), Turkish(tr), Tatar (tt), and Mandarin (zh-TW)*

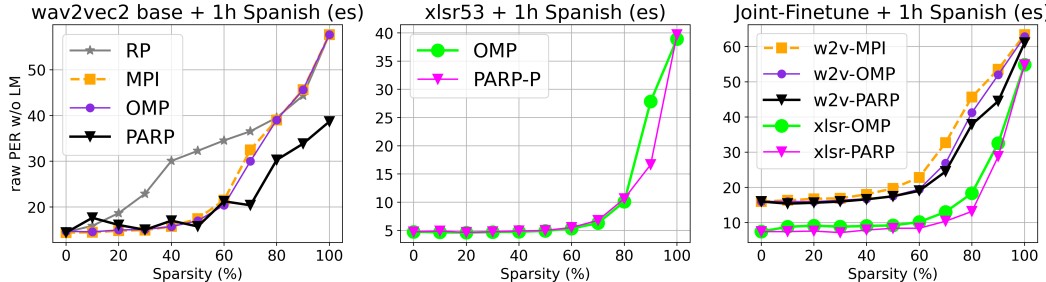

Figure 31: Comparison of pruning techniques on H2L & CSR with 1h of Spanish (*es*) ASR finetuning. **(Left)** Pruning H2L (`wav2vec2-base` + *es*). **(Center)** Pruning CSR (`xlsr` + *es*). **(Right)** Pruning jointly-finetuned `wav2vec2-base` and `xlsr` on *es*.

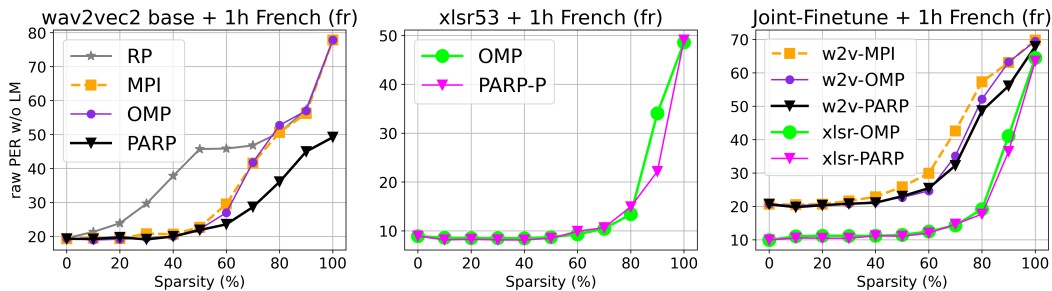

Figure 32: Comparison of pruning techniques on H2L & CSR with 1h of French (*fr*) ASR finetuning. **(Left)** Pruning H2L (`wav2vec2-base` + *fr*). **(Center)** Pruning CSR (`xlsr` + *fr*). **(Right)** Pruning jointly-finetuned `wav2vec2-base` and `xlsr` on *fr*.

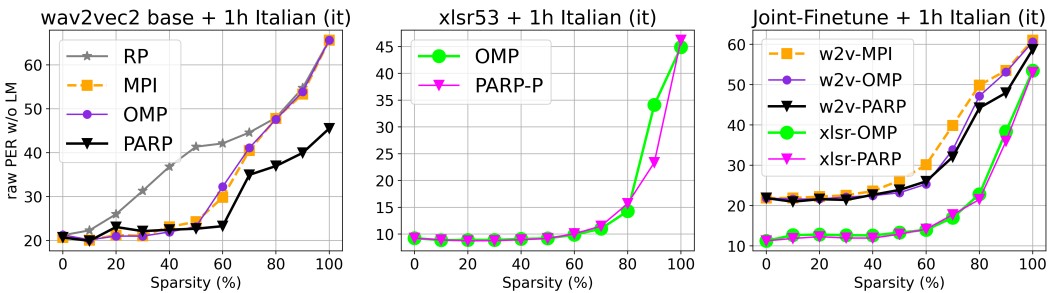

Figure 33: Comparison of pruning techniques on H2L & CSR with 1h of Italian (*it*) ASR finetuning. **(Left)** Pruning H2L (`wav2vec2-base` + *it*). **(Center)** Pruning CSR (`xlsr` + *it*). **(Right)** Pruning jointly-finetuned `wav2vec2-base` and `xlsr` on *it*.

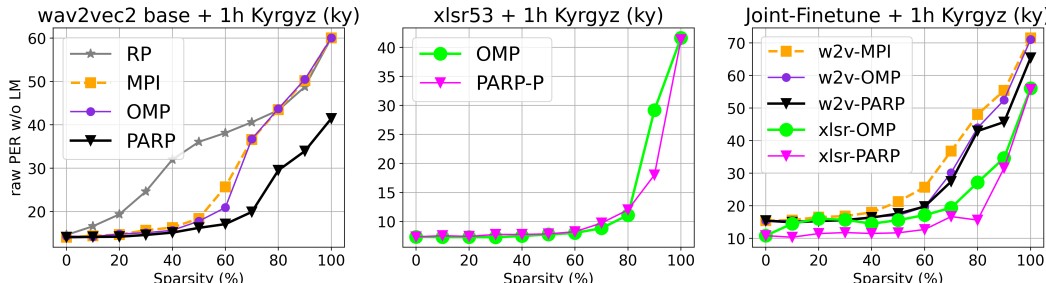

Figure 34: Comparison of pruning techniques on H2L & CSR with 1h of Kyrgyz (*ky*) ASR finetuning. **(Left)** Pruning H2L (`wav2vec2-base` + *ky*). **(Center)** Pruning CSR (`xlsr` + *ky*). **(Right)** Pruning jointly-finetuned `wav2vec2-base` and `xlsr` on *ky*.

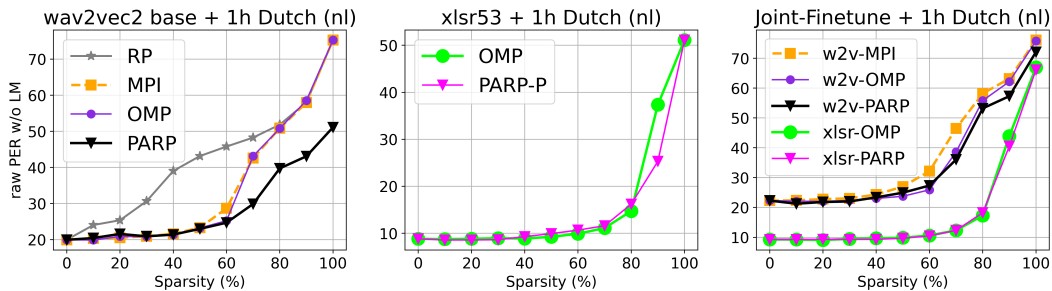

Figure 35: Comparison of pruning techniques on H2L & CSR with 1h of Dutch (*nl*) ASR finetuning. **(Left)** Pruning H2L (`wav2vec2-base` + *nl*). **(Center)** Pruning CSR (`xlsr` + *nl*). **(Right)** Pruning jointly-finetuned `wav2vec2-base` and `xlsr` on *nl*.

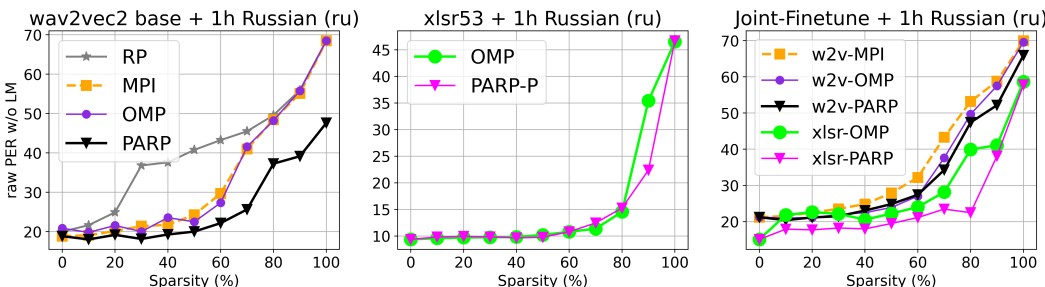

Figure 36: Comparison of pruning techniques on H2L & CSR with 1h of Russian (*ru*) ASR finetuning. **(Left)** Pruning H2L (`wav2vec2-base` + *ru*). **(Center)** Pruning CSR (`xlsr` + *ru*). **(Right)** Pruning jointly-finetuned `wav2vec2-base` and `xlsr` on *ru*.

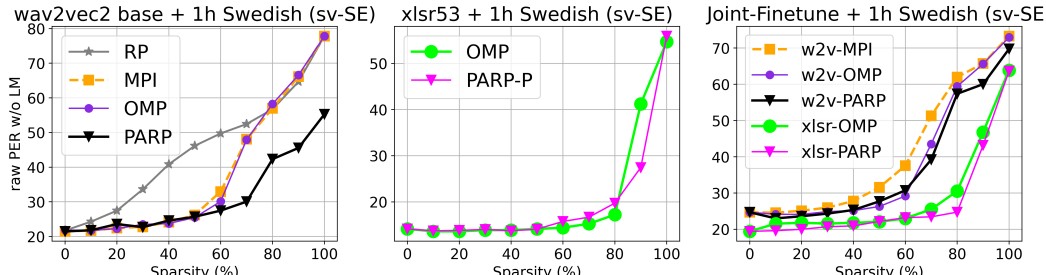

Figure 37: Comparison of pruning techniques on H2L & CSR with 1h of Swedish (*sv-SE*) ASR finetuning. **(Left)** Pruning H2L (`wav2vec2-base` + *sv-SE*). **(Center)** Pruning CSR (`xlsr` + *sv-SE*). **(Right)** Pruning jointly-finetuned `wav2vec2-base` and `xlsr` on *sv-SE*.

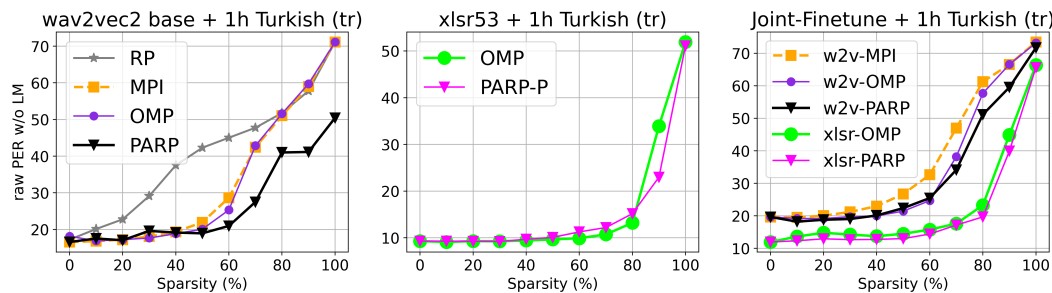

Figure 38: Comparison of pruning techniques on H2L & CSR with 1h of Turkish (*tr*) ASR finetuning. **(Left)** Pruning H2L (`wav2vec2-base` + *tr*). **(Center)** Pruning CSR (`xlsr` + *tr*). **(Right)** Pruning jointly-finetuned `wav2vec2-base` and `xlsr` on *tr*.

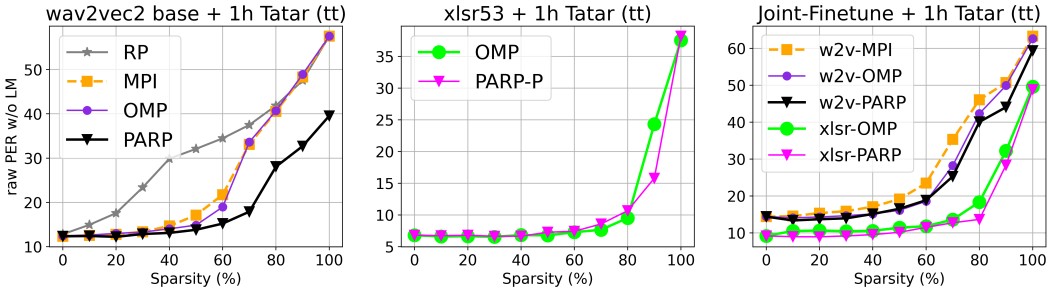

Figure 39: Comparison of pruning techniques on H2L & CSR with 1h of Tatar (*tt*) ASR finetuning. **(Left)** Pruning H2L (`wav2vec2-base` + *tt*). **(Center)** Pruning CSR (`xlsr` + *tt*). **(Right)** Pruning jointly-finetuned `wav2vec2-base` and `xlsr` on *tt*.

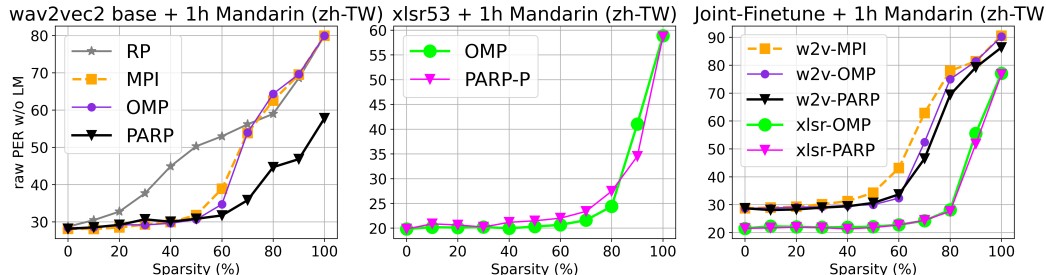

Figure 40: Comparison of pruning techniques on H2L & CSR with 1h of Mandarin (*zh-TW*) ASR finetuning. **(Left)** Pruning H2L (`wav2vec2-base` + *zh-TW*). **(Center)** Pruning CSR (`xlsr` + *zh-TW*). **(Right)** Pruning jointly-finetuned `wav2vec2-base` and `xlsr` on *zh-TW*.

## 15 `wav2vec2` + `PARP` with Random Seeds and LM Decoding

We re-iterate the two reasons why we did not including LM decoding in our main results. First, we isolate the effect of pruning on ASR. Note that the standard LM (either 4-gram/transformer) used in the wav2vec series are also trained on Librispeech (text-corpus) [6, 4]. Therefore, the LMs can easily recover errors made by the acoustic model. Secondly, note that the 4-gram/transformer LM decoding hyper-parameters are carefully searched via Bayesian optimization[14] in the wav2vec series. Such optimization procedure would be quite expensive to run for **just one** model, let alone thousands of pruned models produced in this work.

We provide two sets of results to validate our claim that applying `PARP` on `wav2vec2` reduces the downstream ASR:

- The first result is the impact of random seeds. We finetune `wav2vec2-base` with 10min data at 10% sparsity with `PARP` at 8 additional seeds. Table 5 is the result with Viterbi decoding without LM. We can see that at different seed values, pruned `wav2vec2-base` all converged to similar WERs, which is ∼10% WER reductions compared to a full `wav2vec2-base`.

- The second result is pruned `wav2vec2-base` with the official 4-gram/transformer LM decoding. The pruned `wav2vec-base` is finetuned on the 10min Librispeech split and pruned at 10% sparsity with `PARP`. Since we do not have the compute resource to replicate 1500 beam 4-gram decoding and 500 beam transformer-LM decoding used in the original paper [6], this experiment is based on a more moderate beam size. Similar to [6], decoding hyper-parameters are searched via Ax on the dev-other Librispeech subset over 128 trials. As shown in Table 6, the performance gain over the full `wav2vec2-base` reduces with LM decoding, but we still observe a performance improvement at 10% sparsity with `PARP`.

Table 5: Pruning `wav2vec-base` with `PARP` at different trainnig seeds. Setting is on Librispeech 10min without LM decoding.

| Method | seed | test-clean/test-other |
|---|---|---|
| Full `wav2vec2-base` | 2447 | 49.3/53.2 |
| `wav2vec2-base` + 10% PARP | 2447 | 38.04/44.33 |
| | 0 | 37.01/43.02 |
| | 1 | 37.82/43.66 |
| | 2 | 37.59/43.55 |
| | 3 | 37.57/43.29 |
| | 5 | 37.48/44.10 |
| | 6 | 37.87/43.55 |
| | 7 | 37.65/43.53 |
| | 8 | 38.22/43.91 |

Table 6: Decode pruned `wav2vec-base` with official 4-gram/transformer LMs. Setting is on Librispeech 10min.

| Method | decoding algorithm | beam size | test-clean/test-other |
|---|---|---|---|
| Full `wav2vec2-base` | viterbi (no LM) | | 49.3/53.2 |
| | 4-gram LM | 5 | 27.82/32.02 |
| | transformer LM | 5 | 27.16/32.68 |
| `wav2vec2-base` + 10% PARP averaged | viterbi (no LM) | | 37.69/43.66 |
| | 4-gram LM | 5 | 25.17/32.13 |
| | transformer LM | 5 | 25.45/32.46 |

---

[14]https://github.com/facebook/Ax

# 16 `wav2vec2` Cross-Task Mask Transfer on SUPERB

We extend experiments in Section 4.3 to downstream tasks other than ASR, i.e. extend the transferability of pruning masks across speech tasks. We selected three drastically different target tasks from SUPERB [120]: Phone Recognition with 10h Librispeech data (in PER), Automatic Speaker Verification on VoxCeleb (in EER), and Slot Filling on audio SNIPS (in slot type $F_1$/slot value CER). PER/EER/CER are lower the better, and $F_1$ is higher the better. The experiment procedure [15] is as follows:

1. Finetune `wav2vec2` for a source task in SUPERB.

2. Prune the finetuned model and obtain an `OMP` mask for each task.

3. Apply the `OMP` mask at `wav2vec2` pre-trained initializations and finetune for a target task in SUPERB with `PARP`.

Table 7 is the `wav2vec-base` cross-task transfer result in SUPERB. We did learning rate grid search over $\{1.0 \times 10^{-3}, 1.0 \times 10^{-4}, 1.0 \times 10^{-5}, 2.0 \times 10^{-5}, 3.0 \times 10^{-5}, 1.0 \times 10^{-6}, 1.0 \times 10^{-7}\}$, and presented the best number. Note that different from SUPERB's default setup, we make the upstream `wav2vec2` jointly finetunable for `PARP`. Therefore, the hyper-parameters for each task finetuning are not optimized, and the results here have to be taken with a grain of salt.

Table 7: Cross-task mask transfer for `wav2vec-base` at 50% sparsity.

| Source task | Target task 1: Phone Recog (in PER) | Target task 2: Speaker Verification (in EER) | Target task 3: Slot Filling (in slot type $F_1$/slot value CER) |
|---|---|---|---|
| 10h Librispeech ASR | 0.0567 | 0.1230 | 0.7635/0.4432 |
| 1h Librispeech ASR | 0.0567 | 0.1316 | 0.7563/0.4470 |
| 10min Librispeech ASR | 0.0576 | 0.1399 | 0.7452/0.4596 |
| 10h Phone Recog | 0.0471 | 0.1392 | 0.7575/0.4468 |
| 1h Phone Recog | 0.0483 | 0.1138 | 0.7508/0.4537 |
| 10min Phone Recog | 0.0535 | 0.1224 | 0.7519/0.4596 |
| Intent Classification | 0.0617 | 0.1165 | 0.7490/0.4621 |
| Slot Filling | 0.0601 | 0.1097 | 0.7708/0.4327 |
| Keyword Spotting | 0.0656 | 0.1303 | 0.7490/0.4661 |
| Speaker Verification | 0.0790 | 0.1131 | 0.7497/0.4654 |
| Speaker ID | 0.0677 | 0.1271 | 0.7581/0.4559 |
| Speaker Diarization | 0.0756 | 0.1104 | 0.7449/0.4623 |

We first see that indeed the more similar source and target tasks are, the performance are better. For instance, source subnetwork obtained from speaker related task perform better than those obtained from ASR/keyword spotting on speaker verification. For another, source subnetwork obtained from ASR/phone recognition perform better than those obtained from speaker related task on phone recognition. We do note that the numbers are not off by too much, and the differences could be potentially reduced via hyper-parameter tuning. This pilot study also suggests that subnetworks transferability depends on task similarity. Lastly, this experiment does not contradict our main setting, as we were primarily interested in cross-lingual transferability of subnetworks in Section 4.3.

---

[15] All experiments are run with SUPERB's toolkit `https://github.com/s3prl/s3prl`.

# 17  Does Observation 1 generalize across Pre-Training Objectives?

Observation 1 states that:

> *For any sparsity, any amount of finetuning supervision, any pre-training model scale, and any downstream spoken languages, the non-zero ASR pruning masks obtained from task-agnostic subnetwork discovery has high IOUs with those obtained from task-aware subnetwork discovery.*

We provide analysis on whether Observation 1 holds *across* pre-training objectives, i.e. does pruning masks from `wav2vec2` have high similarity with those from `hubert` [55]? The setup follows that of Section 16 and is based on the downstream tasks in SUPERB[16]:

1. Finetune `wav2vec2` for all tasks in SUPERB.
2. Prune the finetuned models and obtain an `OMP` mask for each task.
3. Finetune `hubert` for all tasks in SUPERB
4. Prune the finetuned models and obtain an `OMP` mask for each task.
5. For each task in SUPERB and at a fixed sparsity, calculate the mask `IOU` between `wav2vec2` and `hubert`.

Table 8 is the mask `IOU`s at 50% sparsity between `wav2vec-base` and `hubert-base` on tasks in SUPERB. The table indicates that while Observation 1 holds separately for `wav2vec2` (contrastive pre-training) and `hubert` (mask-predict pre-training), it does not generalize across pre-training method give the close to random mask `IOU`s (c.f. last row of Table 8). Therefore,

> *Observation 1 holds true conditioned on the same speech SSL pre-training objective.*

Table 8: Mask `IOU` between `wav2vec-base` and `hubert-base` at 50% sparsity.

| target task | mask IOU between `wav2vec-base` and `hubert-base` |
|---|---|
| 10h Librispeech ASR | 0.3472 |
| 1h Librispeech ASR | 0.3473 |
| 10min Librispeech ASR | 0.3473 |
| 10h Phone Recog | 0.3473 |
| 1h Phone Recog | 0.3473 |
| 10min Phone Recog | 0.3473 |
| Intent Classification | 0.3473 |
| Slot Filling | 0.3472 |
| Keyword Spotting | 0.3473 |
| Speaker Verification | 0.3473 |
| Speaker ID | 0.3473 |
| Speaker Diarization | 0.3472 |
| Random Pruning | 0.3473 |

This finding is perhaps not so surprising, see prior work on similarity analysis between contextualized speech [24] and word [113] representations. They suggest that different pre-trained models' contextualized representations have low similarities, e.g. BERT v.s. XLNet. We stress that this does not invalidate PARP. As long as Observation 1 holds, PARP's step 2 should make learnable adjustments to the initial mask given the high overlaps between pruning masks.

---

[16]For this set of experiments, we used the same optimization method (Adam with constant $1.0 \times 10^{-5}$ learning rate) for finetuning `wav2vec-base` and `hubert-base`.

## 18 Pruned Weights Localization Across Layers

The wav2vec series [6, 4, 55, 53] is known to have more valuable contextualized representations towards the middle of the network for downstream ASR. We examine whether previous observations holds true for pruning, that weights in middle layers are pruned less. To understand such a phenomenon, we calculated the distributions of the pruned weights/neurons across each layer, and an example is shown in Table 9.

Table 9: `wav2vec-base` finetuned for Spanish (**H2L** setting) pruned at 50% sparsity with `OMP`.

| layer | 1 | 2 | 3 | 4 | 5 | 6 | 7 | 8 | 9 | 10 | 11 | 12 |
|---|---|---|---|---|---|---|---|---|---|---|---|---|
| sparsity (%) | 53.52 | 52.45 | 49.24 | 47.90 | 46.51 | 46.84 | 45.97 | 45.58 | 45.96 | 47.96 | 52.54 | 65.53 |

Table 9 shows that indeed bottom and higher layers of `wav2vec2-base` are pruned more, while the middle layers are pruned less. We observe similar pruned weight distributions across spoken languages (10 languages) and sparsities (10%, 20%, 30%, . . . , 90%). See the rest of the sparsity distribution in the Figures below. This analysis suggests that regardless of spoken languages, intermediate layers' neurons are more valuable than lower and higher-level layers, manifested by the layer's sparsity ratio.

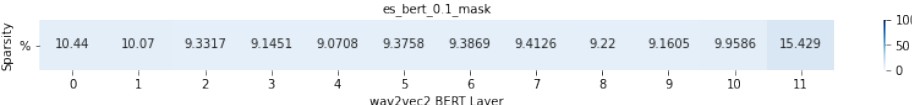

Figure 41: Sparsity over layers for `wav2vec-base` finetuned for Spanish *es* at 10% sparsity.

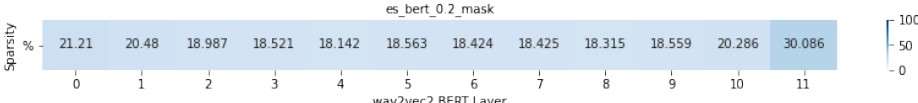

Figure 42: Sparsity over layers for `wav2vec-base` finetuned for Spanish *es* at 20% sparsity.

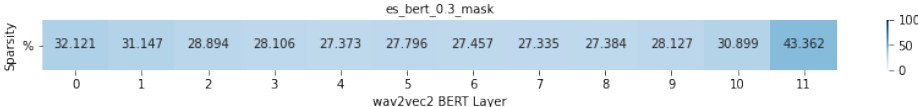

Figure 43: Sparsity over layers for `wav2vec-base` finetuned for Spanish *es* at 30% sparsity.

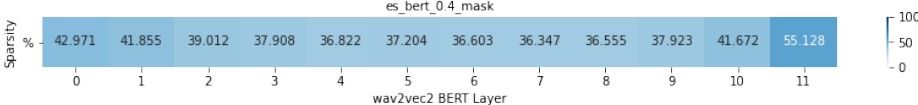

Figure 44: Sparsity over layers for `wav2vec-base` finetuned for Spanish *es* at 40% sparsity.

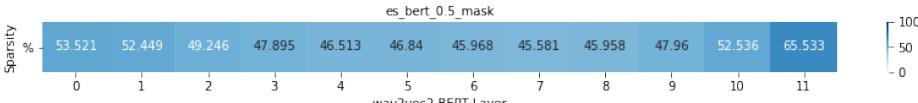

Figure 45: Sparsity over layers for `wav2vec-base` finetuned for Spanish *es* at 50% sparsity.

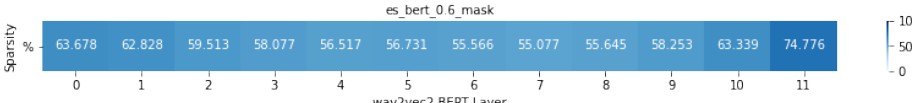

Figure 46: Sparsity over layers for `wav2vec-base` finetuned for Spanish *es* at 60% sparsity.

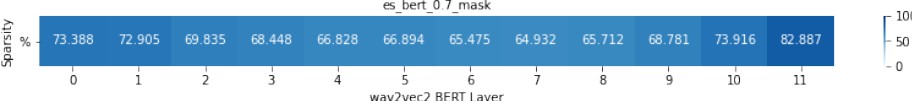

Figure 47: Sparsity over layers for `wav2vec-base` finetuned for Spanish *es* at 70% sparsity.

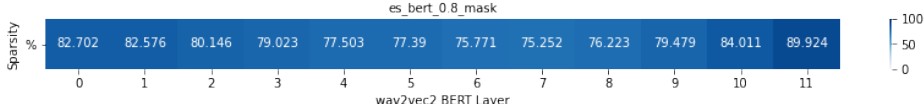

Figure 48: Sparsity over layers for `wav2vec-base` finetuned for Spanish *es* at 80% sparsity.

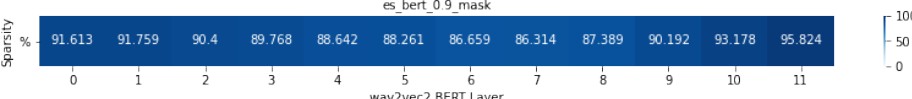

Figure 49: Sparsity over layers for `wav2vec-base` finetuned for Spanish *es* at 90% sparsity.

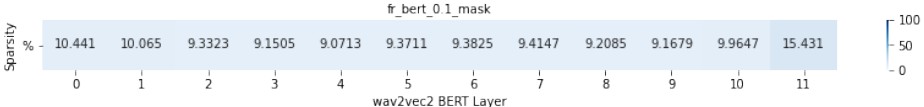

Figure 50: Sparsity over layers for `wav2vec-base` finetuned for French *fr* at 10% sparsity.

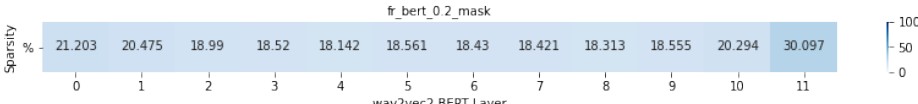

Figure 51: Sparsity over layers for `wav2vec-base` finetuned for French *fr* at 20% sparsity.

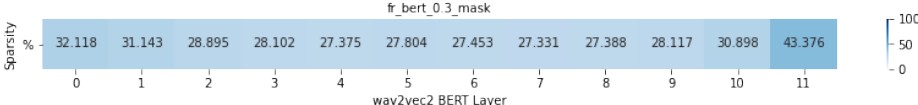

Figure 52: Sparsity over layers for `wav2vec-base` finetuned for French *fr* at 30% sparsity.

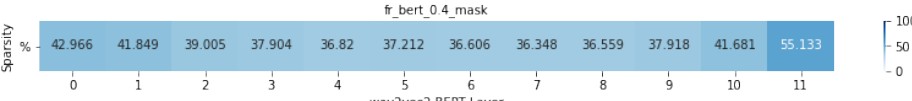

Figure 53: Sparsity over layers for `wav2vec-base` finetuned for French *fr* at 40% sparsity.

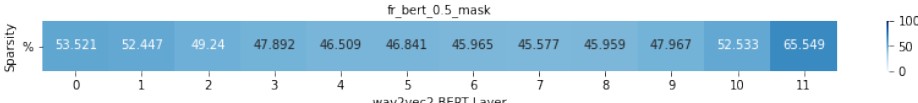

Figure 54: Sparsity over layers for `wav2vec-base` finetuned for French *fr* at 50% sparsity.

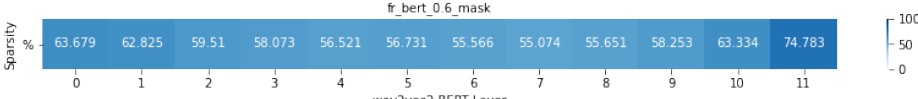

Figure 55: Sparsity over layers for `wav2vec-base` finetuned for French *fr* at 60% sparsity.

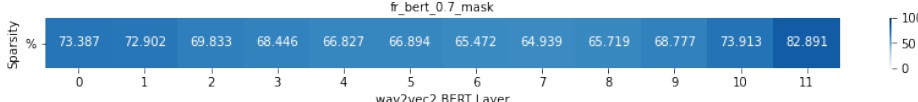

Figure 56: Sparsity over layers for `wav2vec-base` finetuned for French *fr* at 70% sparsity.

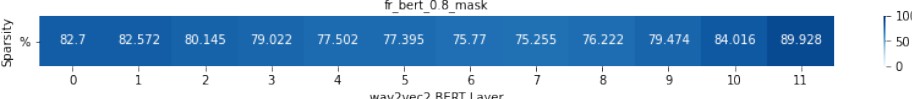

Figure 57: Sparsity over layers for `wav2vec-base` finetuned for French *fr* at 80% sparsity.

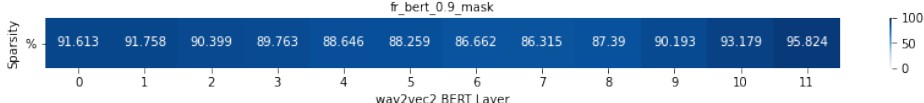

Figure 58: Sparsity over layers for `wav2vec-base` finetuned for French *fr* at 90% sparsity.

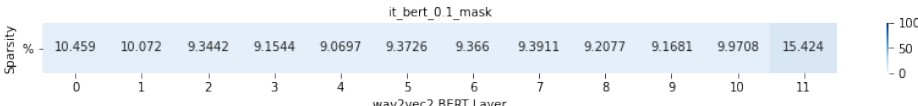

Figure 59: Sparsity over layers for `wav2vec-base` finetuned for Italian *it* at 10% sparsity.

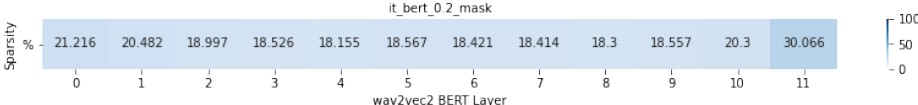

Figure 60: Sparsity over layers for `wav2vec-base` finetuned for Italian *it* at 20% sparsity.

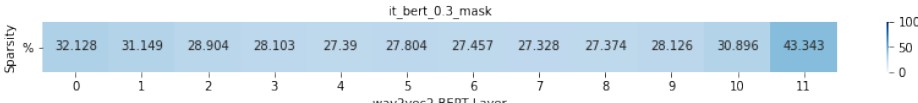

Figure 61: Sparsity over layers for `wav2vec-base` finetuned for Italian *it* at 30% sparsity.

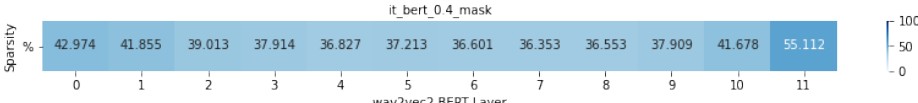

Figure 62: Sparsity over layers for `wav2vec-base` finetuned for Italian *it* at 40% sparsity.

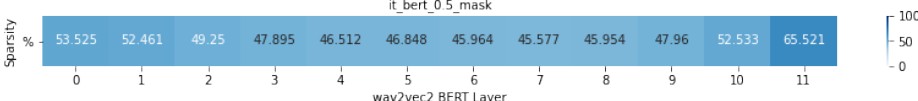

Figure 63: Sparsity over layers for `wav2vec-base` finetuned for Italian *it* at 50% sparsity.

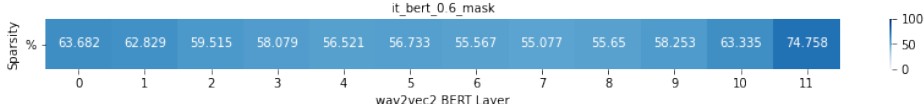

Figure 64: Sparsity over layers for `wav2vec-base` finetuned for Italian *it* at 60% sparsity.

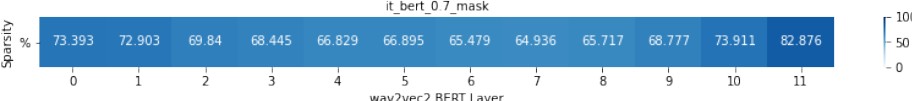

Figure 65: Sparsity over layers for `wav2vec-base` finetuned for Italian *it* at 70% sparsity.

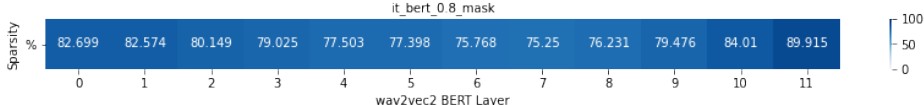

Figure 66: Sparsity over layers for `wav2vec-base` finetuned for Italian *it* at 80% sparsity.

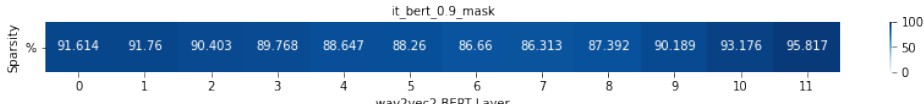

Figure 67: Sparsity over layers for `wav2vec-base` finetuned for Italian *it* at 90% sparsity.

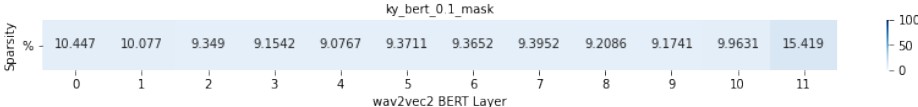

Figure 68: Sparsity over layers for `wav2vec-base` finetuned for Kyrgyz *ky* at 10% sparsity.

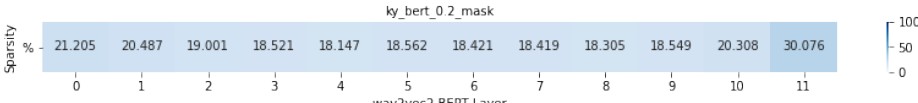

Figure 69: Sparsity over layers for `wav2vec-base` finetuned for Kyrgyz *ky* at 20% sparsity.

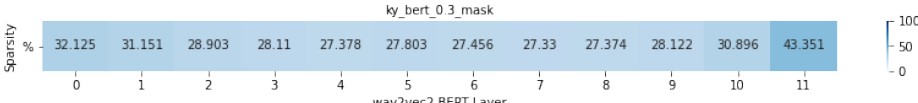

Figure 70: Sparsity over layers for `wav2vec-base` finetuned for Kyrgyz *ky* at 30% sparsity.

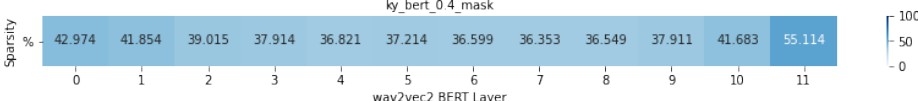

Figure 71: Sparsity over layers for `wav2vec-base` finetuned for Kyrgyz *ky* at 40% sparsity.

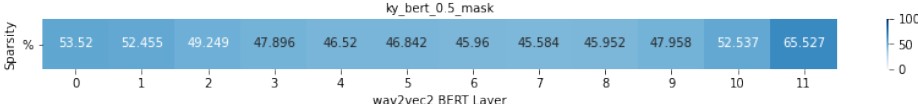

Figure 72: Sparsity over layers for `wav2vec-base` finetuned for Kyrgyz *ky* at 50% sparsity.

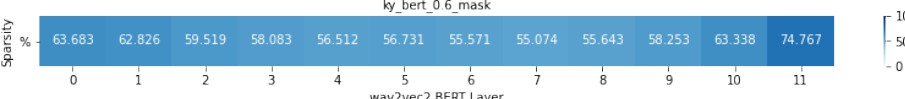

Figure 73: Sparsity over layers for `wav2vec-base` finetuned for Kyrgyz *ky* at 60% sparsity.

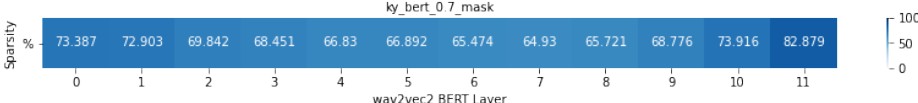

Figure 74: Sparsity over layers for `wav2vec-base` finetuned for Kyrgyz *ky* at 70% sparsity.

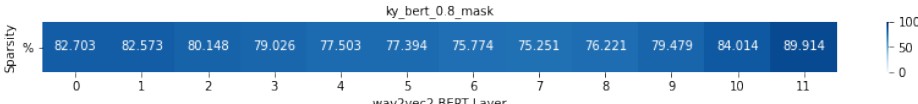

Figure 75: Sparsity over layers for `wav2vec-base` finetuned for Kyrgyz *ky* at 80% sparsity.

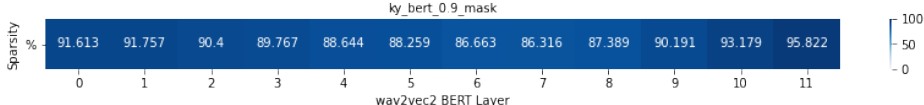

Figure 76: Sparsity over layers for `wav2vec-base` finetuned for Kyrgyz *ky* at 90% sparsity.

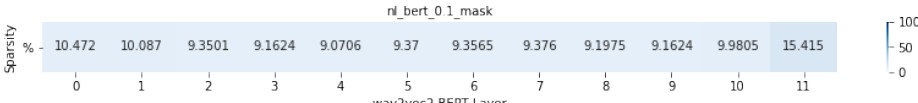

Figure 77: Sparsity over layers for `wav2vec-base` finetuned for Dutch *nl* at 10% sparsity.

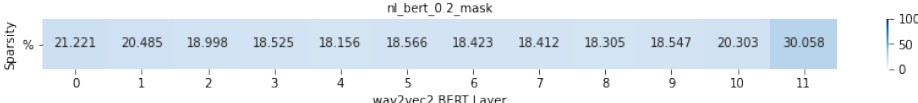

Figure 78: Sparsity over layers for `wav2vec-base` finetuned for Dutch *nl* at 20% sparsity.

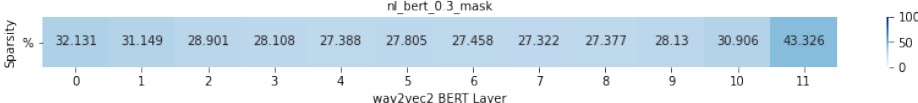

Figure 79: Sparsity over layers for `wav2vec-base` finetuned for Dutch *nl* at 30% sparsity.

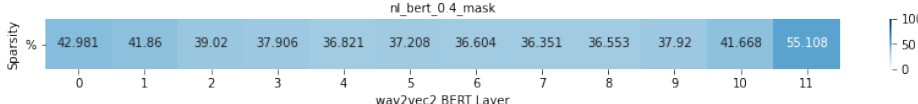

Figure 80: Sparsity over layers for `wav2vec-base` finetuned for Dutch *nl* at 40% sparsity.

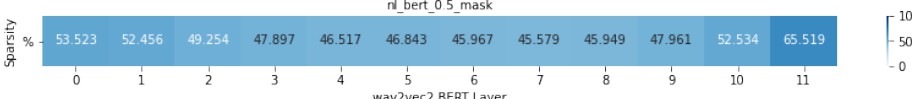

Figure 81: Sparsity over layers for `wav2vec-base` finetuned for Dutch *nl* at 50% sparsity.

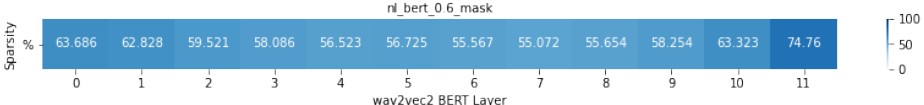

Figure 82: Sparsity over layers for `wav2vec-base` finetuned for Dutch *nl* at 60% sparsity.

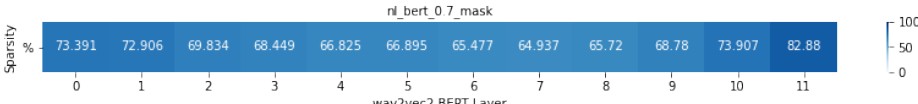

Figure 83: Sparsity over layers for `wav2vec-base` finetuned for Dutch *nl* at 70% sparsity.

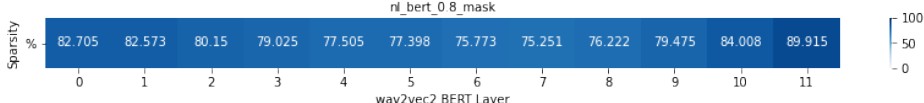

Figure 84: Sparsity over layers for `wav2vec-base` finetuned for Dutch *nl* at 80% sparsity.

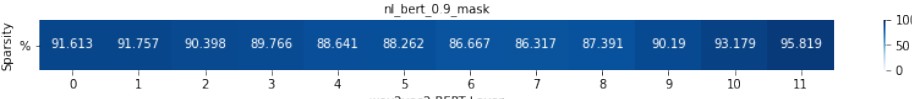

Figure 85: Sparsity over layers for `wav2vec-base` finetuned for Dutch *nl* at 90% sparsity.

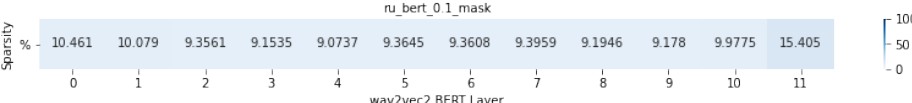

Figure 86: Sparsity over layers for `wav2vec-base` finetuned for Russian *ru* at 10% sparsity.

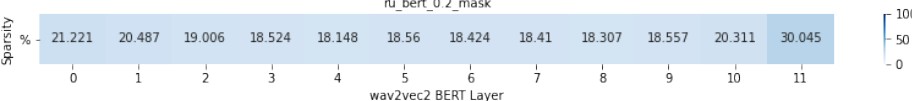

Figure 87: Sparsity over layers for `wav2vec-base` finetuned for Russian *ru* at 20% sparsity.

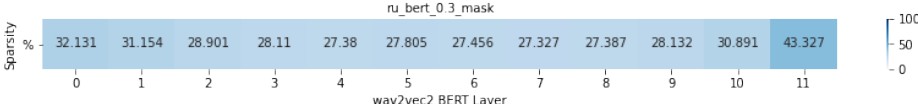

Figure 88: Sparsity over layers for `wav2vec-base` finetuned for Russian *ru* at 30% sparsity.

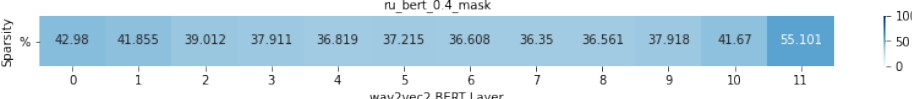

Figure 89: Sparsity over layers for `wav2vec-base` finetuned for Russian *ru* at 40% sparsity.

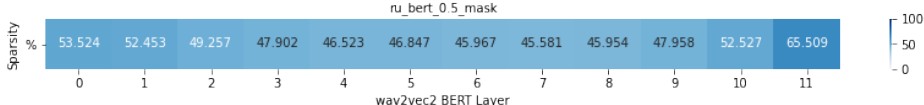

Figure 90: Sparsity over layers for `wav2vec-base` finetuned for Russian *ru* at 50% sparsity.

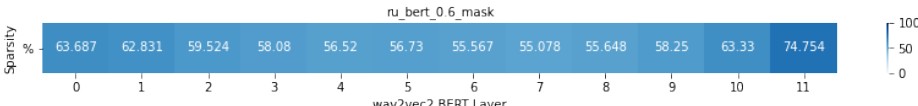

Figure 91: Sparsity over layers for `wav2vec-base` finetuned for Russian *ru* at 60% sparsity.

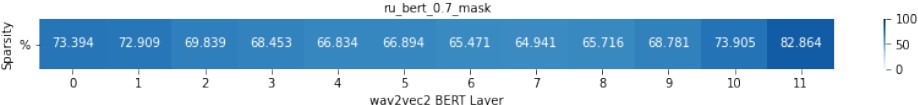

Figure 92: Sparsity over layers for `wav2vec-base` finetuned for Russian *ru* at 70% sparsity.

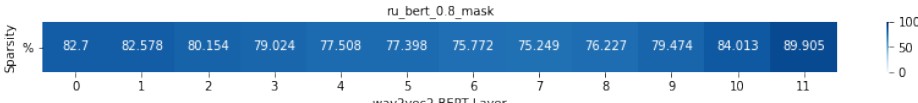

Figure 93: Sparsity over layers for `wav2vec-base` finetuned for Russian *ru* at 80% sparsity.

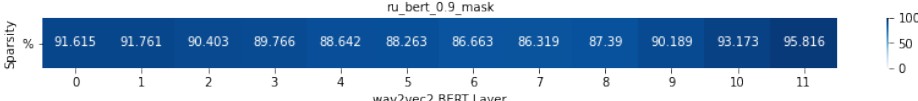

Figure 94: Sparsity over layers for `wav2vec-base` finetuned for Russian *ru* at 90% sparsity.

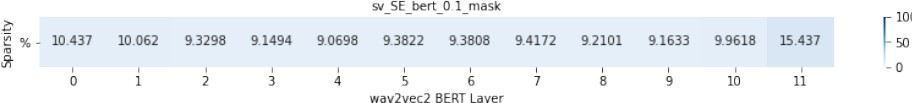

Figure 95: Sparsity over layers for `wav2vec-base` finetuned for Swedish *sv-SE* at 10% sparsity.

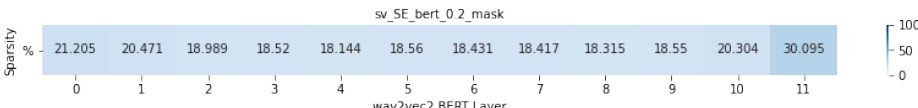

Figure 96: Sparsity over layers for `wav2vec-base` finetuned for Swedish *sv-SE* at 20% sparsity.

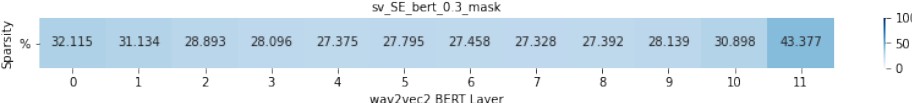

Figure 97: Sparsity over layers for `wav2vec-base` finetuned for Swedish *sv-SE* at 30% sparsity.

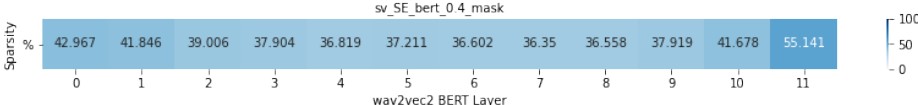

Figure 98: Sparsity over layers for `wav2vec-base` finetuned for Swedish *sv-SE* at 40% sparsity.

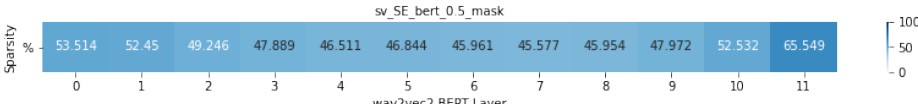

Figure 99: Sparsity over layers for `wav2vec-base` finetuned for Swedish *sv-SE* at 50% sparsity.

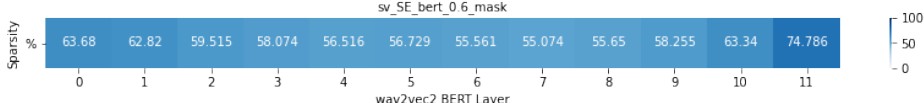

Figure 100: Sparsity over layers for `wav2vec-base` finetuned for Swedish *sv-SE* at 60% sparsity.

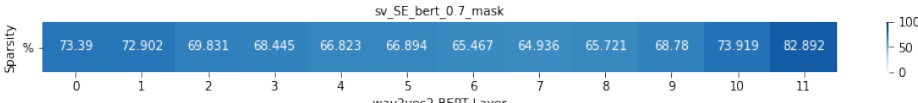

Figure 101: Sparsity over layers for `wav2vec-base` finetuned for Swedish *sv-SE* at 70% sparsity.

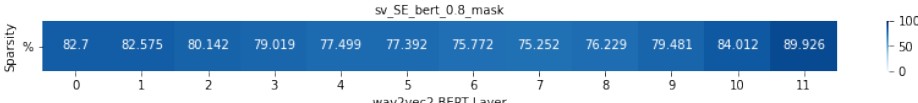

Figure 102: Sparsity over layers for `wav2vec-base` finetuned for Swedish *sv-SE* at 80% sparsity.

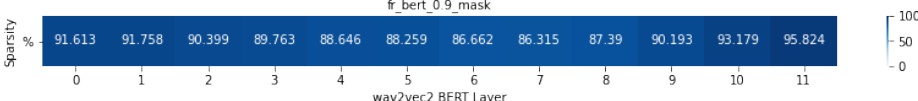

Figure 103: Sparsity over layers for `wav2vec-base` finetuned for Swedish *sv-SE* at 90% sparsity.

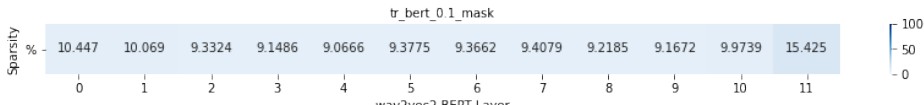

Figure 104: Sparsity over layers for `wav2vec-base` finetuned for Turkish *tr* at 10% sparsity.

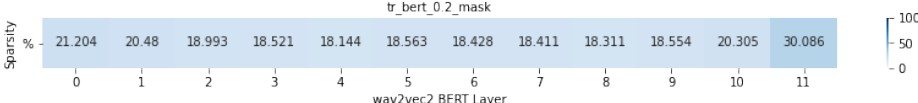

Figure 105: Sparsity over layers for `wav2vec-base` finetuned for Turkish *tr* at 20% sparsity.

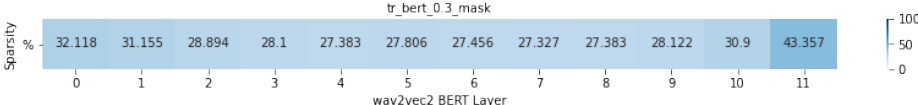

Figure 106: Sparsity over layers for `wav2vec-base` finetuned for Turkish *tr* at 30% sparsity.

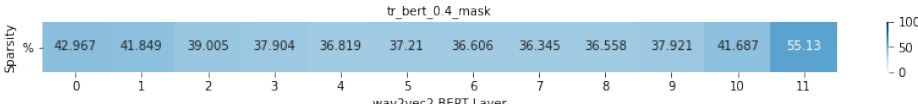

Figure 107: Sparsity over layers for `wav2vec-base` finetuned for Turkish *tr* at 40% sparsity.

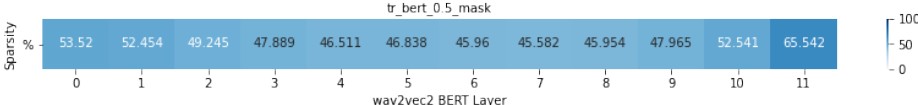

Figure 108: Sparsity over layers for `wav2vec-base` finetuned for Turkish *tr* at 50% sparsity.

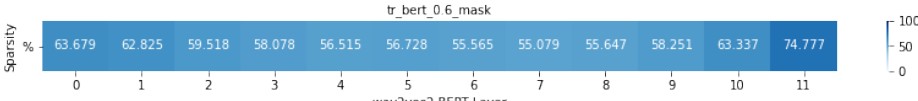

Figure 109: Sparsity over layers for `wav2vec-base` finetuned for Turkish *tr* at 60% sparsity.

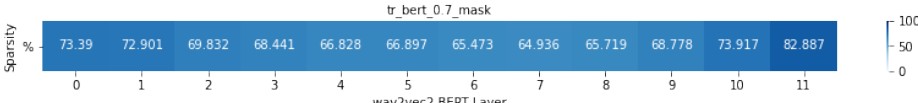

Figure 110: Sparsity over layers for `wav2vec-base` finetuned for Turkish *tr* at 70% sparsity.

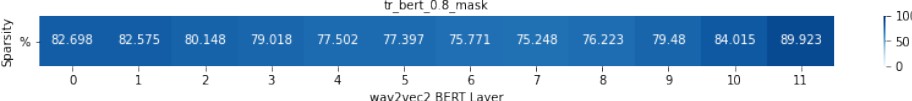

Figure 111: Sparsity over layers for `wav2vec-base` finetuned for Turkish *tr* at 80% sparsity.

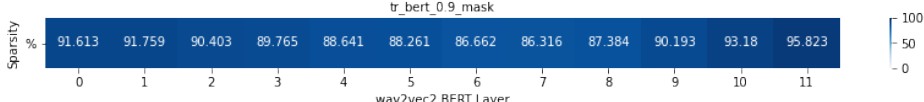

Figure 112: Sparsity over layers for `wav2vec-base` finetuned for Turkish *tr* at 90% sparsity.

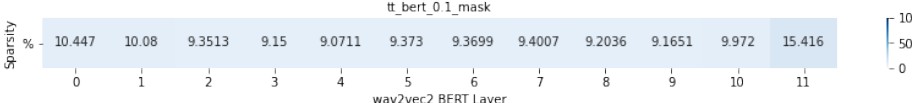

Figure 113: Sparsity over layers for `wav2vec-base` finetuned for Tatar *tt* at 10% sparsity.

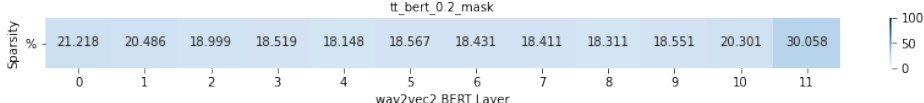

Figure 114: Sparsity over layers for `wav2vec-base` finetuned for Tatar *tt* at 20% sparsity.

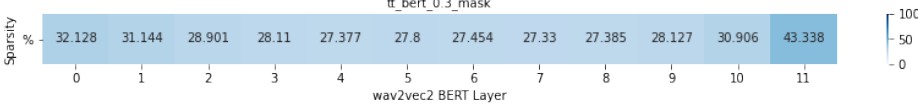

Figure 115: Sparsity over layers for `wav2vec-base` finetuned for Tatar *tt* at 30% sparsity.

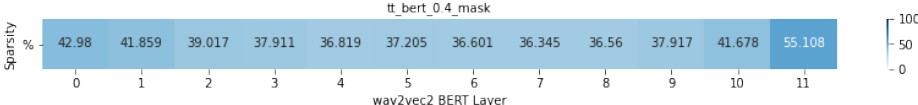

Figure 116: Sparsity over layers for `wav2vec-base` finetuned for Tatar *tt* at 40% sparsity.

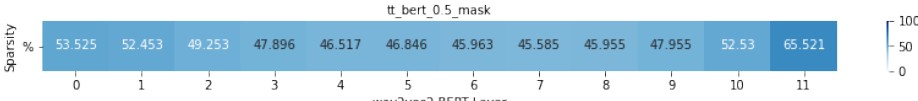

Figure 117: Sparsity over layers for `wav2vec-base` finetuned for Tatar *tt* at 50% sparsity.

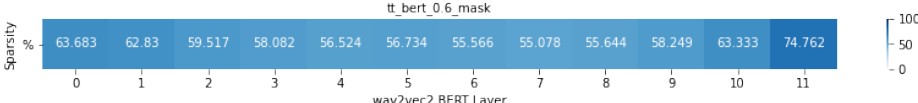

Figure 118: Sparsity over layers for `wav2vec-base` finetuned for Tatar *tt* at 60% sparsity.

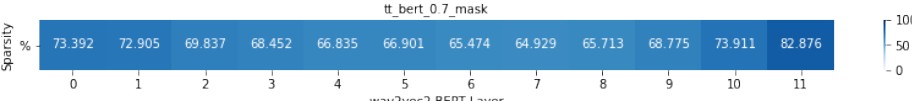

Figure 119: Sparsity over layers for `wav2vec-base` finetuned for Tatar *tt* at 70% sparsity.

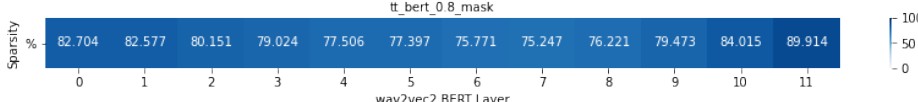

Figure 120: Sparsity over layers for `wav2vec-base` finetuned for Tatar *tt* at 80% sparsity.

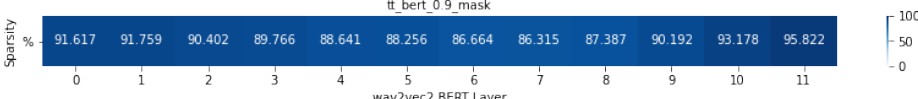

Figure 121: Sparsity over layers for `wav2vec-base` finetuned for Tatar *tt* at 90% sparsity.

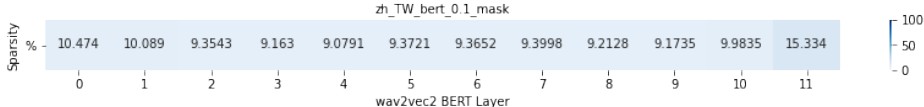

Figure 122: Sparsity over layers for `wav2vec-base` finetuned for Mandarin *zh-TW* at 10% sparsity.

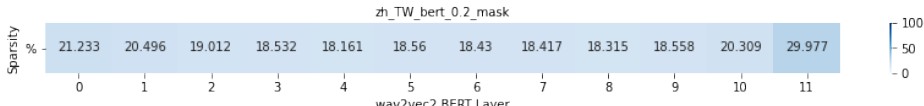

Figure 123: Sparsity over layers for `wav2vec-base` finetuned for Mandarin *zh-TW* at 20% sparsity.

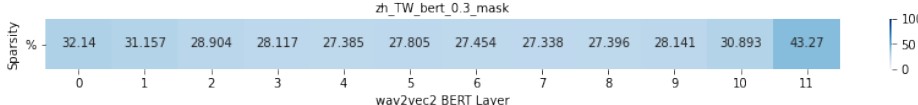

Figure 124: Sparsity over layers for `wav2vec-base` finetuned for Mandarin *zh-TW* at 30% sparsity.

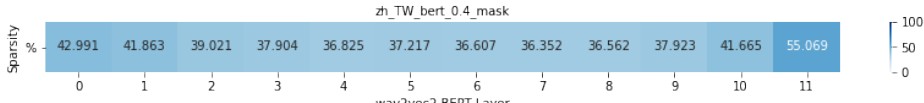

Figure 125: Sparsity over layers for `wav2vec-base` finetuned for Mandarin *zh-TW* at 40% sparsity.

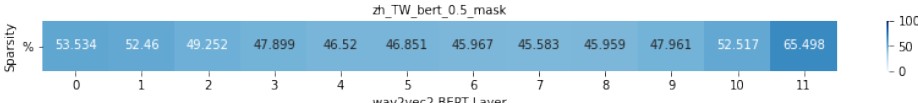

Figure 126: Sparsity over layers for `wav2vec-base` finetuned for Mandarin *zh-TW* at 50% sparsity.

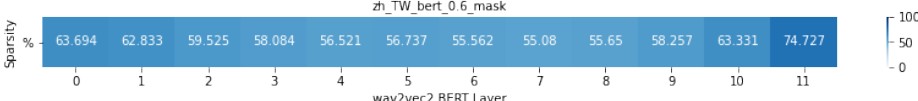

Figure 127: Sparsity over layers for `wav2vec-base` finetuned for Mandarin *zh-TW* at 60% sparsity.

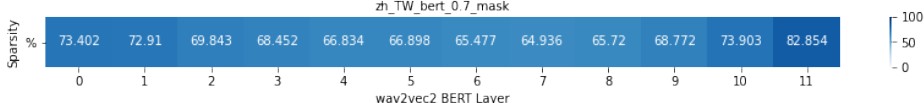

Figure 128: Sparsity over layers for `wav2vec-base` finetuned for Mandarin *zh-TW* at 70% sparsity.

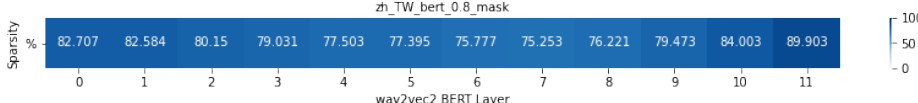

Figure 129: Sparsity over layers for `wav2vec-base` finetuned for Mandarin *zh-TW* at 80% sparsity.

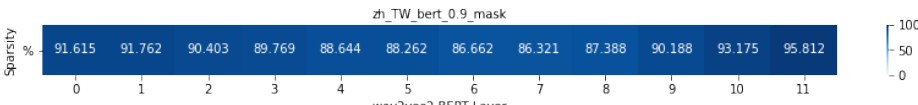

Figure 130: Sparsity over layers for `wav2vec-base` finetuned for Mandarin *zh-TW* at 90% sparsity.

# 19 Experimental Limitations

Below, we list several limiting factors of our experimental designs:

1. Experiments are on contrastive pre-trained models only. It is unclear whether the results would generalize to pre-trained models with other objectives, such as mask prediction (HuBERT) or autoregressive prediction (APC), etc.

2. Although standard, our experiments are on relatively large pre-trained models (number of parameter is 90M for `wav2vec2-base` and 315M for `wav2vec2-large` and `xlsr`. It would be interesting to investigate if small pre-trained models can also be pruned and whether Observation 1 holds for them.

3. Our `wav2vec2-base` and `wav2vec2-large` are both pre-trained on Librispeech 960 hours. Another lack of study is the effect of pre-training data selections – what happens if pre-training and fine-tuning data are from different sources?

4. Our fine-tuning dataset (Librispeech and CommonVoice) are both read speech. Experiments on conversational (e.g. telephone) speech should be investigated.

5. In addition, though opposite to our motivation, it is unclear is the results hold for high-resource languages (e.g. 100h∼1000h of fine-tuning data).

6. Our ASR experiments are based on self-supervised pre-trained models. It remains to be studied on applying PARP to E2E ASR without self-supervised pre-training.

7. Lastly, we note that this study is scientific by nature. Observation 1 emerges after our initial pilot study, and it motivates the central idea of PARP. We will leave it to follow-up work to test whether such pruning method is effective in more realistic settings (e.g. noisy data, limited bandwidth, etc).