# OpenReview forum: "PARP: Prune, Adjust and Re-Prune for Self-Supervised Speech Recognition"
_NeurIPS.cc/2021/Conference — NeurIPS 2021 Spotlight_

### Official Review · Reviewer_hRhS · 2021-07-16

**Rating:** 8
**Confidence:** 4

**Summary:**

In this work, the authors build upon techniques to reduce the number of parameters in the model by finding sparse sub-networks along the lines of the “lottery-ticket hypothesis” work. In this case, however, the authors apply the proposed techniques in the context of self-supervised speech recognition by showing that the same initial masked network is suitable for fine-tuning for a number of tasks including when applied for cross-lingual transfer when used with multiple languages.

**Limitations And Societal Impact:**

Apart from the limitations mentioned above, I do not see any potential negative societal impacts of this work.

**Main Review:**

Overall, this is an extremely well-written paper. In particular, I found that the motivation of the work was extremely clear, and well-supported by analyses to show that the sub-networks obtained in a task agnostic fashion had a large overlap (as measured in terms of IOU) with the task-specific sub-networks. I also found the experiments to be well-designed and comprehensive exploring different settings: pre-training and finetuning for the same language; pre-training on one language and fine-tuning on a large number of languages; pre-training and fine-tuning on multiple languages. The ablation experiments in Section 4, were interesting, and strengthen the paper greatly. In my view, the authors’ work -- particularly, the work on rapidly training models on multiple languages -- is a very impactful contribution.

One clarification question: In Figure 4, if I understand correctly, the PER w/o and LM is still ~50% even at 100% sparsity. This was somewhat puzzling to me. Shouldn’t the PER be 100%+ if all weights are zeroed-out? (I assume that 100% sparsity means that all weights are zeroed out even after finetuning).

I would note that one limitation of this work is that although the authors show that they can achieve large unstructured sparsity in the model, it is not clear if the sparsity structure will be amenable to getting large speed-ups when run on a device/server. I think the paper could be strengthened by measuring actual RTF/benchmark numbers by running these models on device.


**Time Spent Reviewing:**

4

---

> ### Author Response · Authors · 2021-08-10
> **Response to Reviewer hRhS**
>
> *Q1: One clarification question: In Figure 4, if I understand correctly, the PER w/o and LM is still ~50% even at 100% sparsity. This was somewhat puzzling to me. Shouldn’t the PER be 100%+ if all weights are zeroed-out? (I assume that 100% sparsity means that all weights are zeroed out even after finetuning).*
>
> We thank the reviewer for the comments.
>
> Regarding the PER at 100% sparsity, this is due to our decision of not pruning (1) the CNN feature extractor in wav2vec2/XLSR53, (2) layer norm running statistics within wav2vec2/XLSR53, and (3) task-specific linear layer. For (1), it is due to the CNN feature extractor being fixed during finetuning by default, and the majority of the model parameters lie in wav2vec2/XLSR53’s BERT. For (2)(3), we simply follow the setup described in BERT-Ticket. In fact, while re-producing BERT-Ticket, we were also surprised that BERT’s layer norm itself plus its final linear layer achieve non-trivial loss/accuracy (e.g. BERT’s MLM at 0% sparsity is ~60% acc while at 100% sparsity is ~15% acc.). However, it was not immediately clear to us how running statistics and task-specific linear layers can be pruned.
>
> *Q2: I would note that one limitation of this work is that although the authors show that they can achieve large unstructured sparsity in the model, it is not clear if the sparsity structure will be amenable to getting large speed-ups when run on a device/server. I think the paper could be strengthened by measuring actual RTF/benchmark numbers by running these models on device.*
>
> Regarding benchmarking models on edge devices, we fully agree and are also curious if PARP provides inference speedup in realistic settings (in fact, this is one of the limitations we listed in Appendix H). However, we, unfortunately, do not have access to mobile devices. Even recent ASR pruning papers from industry teams lack reports of inference speedup, see [1,2]. One of the main reasons could be attributed to the diverse conditions of speech datasets (some datasets like Youtube have >10min utterances, while some have 1-5 second short utterances like Speech Commands), and therefore it is difficult to fairly compare pruning methods from different papers in terms of inference speed.
>
> Furthermore, we would like to stress that this work is scientific in nature, and we are driven by questions: how sparse are recent speech SSL models? Are there any peculiarities in their pruning patterns? We selected unstructured magnitude pruning as the basis for our investigation and for PARP, since it is the most flexible and widely adopted pruning mechanism in literature, see tutorials like [3] and all lottery ticket hypothesis papers.
>
> Reference: \
> [1] https://arxiv.org/pdf/1909.12408.pdf \
> [2] https://arxiv.org/pdf/2005.10627.pdf \
> [3] https://pytorch.org/tutorials/intermediate/pruning_tutorial.html

---

> > ### Comment · Reviewer_hRhS · 2021-08-19
> > **Thanks for the detailed responses to the reviewer comments**
> >
> > I would like to thank the authors for their detailed replies to all of the reviewers' comments. As I mentioned in my first review, I think this is a great paper, and I would like to retain my previous score.

---

### Official Review · Reviewer_xQYW · 2021-08-04

**Rating:** 7
**Confidence:** 3

**Summary:**

This paper presents a Prune-Adjust-Re-Prune (PARP) approach for finding a sparse subnetwork within a self-supervised speech model with competitive performance for the Automatic Speech Recognition (ASR) downstream task in mono-lingual and multi-lingual conditions. Despite the approach's simplicity, its downstream ASR performance (measured using WER without LM decoding) is better than other subnetwork discovery approaches (for almost all fine-tuning conditions) and the original unpruned network when 10mins of paired data is used.


**Limitations And Societal Impact:**

yes

**Main Review:**

The proposed PARP approach alternates between two steps starting from the pre-trained network: a) zeroing out the smallest X % of the weights based on their magnitude. b) training the network for some update steps while allowing the zeroed-out weights in step (a) to grow again. The proposed approach is similar to previous work in NN pruning, but it is the first to be applied to self-supervised speech models. Experiments on multi-lingual ASR showed that initial subnetwork masks could be transferred between languages. Also, a quick experiment on NLP tasks showed similar behavior. The paper is well written with good experiments on the Wav2vec 2.0 pre-trained English and other lower-resource languages.

There are few comments on the presented work:
1) The paper offers an imbalanced presentation between Speech and NLP. Given the generality of the proposed approach and continuous reference to previously proposed work on other modalities, it makes sense to offer more experiments on the same tasks. Adding Table 4 raises the confidence in the efficacy of the approach a bit. More experiments will convince the reader further.

2) Using WER with no LM is accepted for seq2seq models which learn an implicit LM from the data; however, this work uses only a CTC loss leading to considerable variance in the WER when no LM is used, in particular when few amounts of paired data are used, e.g., 10mins. The authors are encouraged to share results with n-gram decoding, at least for their English models.

3) The results on initial subnetwork generality to multiple tasks for the speech case need to be taken cautiously since all the other tasks are still ASR (but in other languages). The bottom part of many models trained from waveform input captures spectral representations shared between languages for the same task. Audio Event Detection tasks or Speaker Clustering tasks may be good ones to examine this point for speech.

4) There is not much discussion on where the pruned weights are located in the pre-trained network. The Wav2vec2.0 approach is known to have valuable representations towards the middle of the network, with higher-level weights only beneficial for the pre-training contrastive loss with less impact on downstream applications. One reason for the similarity between the subnetwork for English and multi-lingual models could be that both use the same pre-training approach, leading to the same inductive biases, i.e., a peculiarity of the pre-training approach rather than a property of the found sub-network used with PARP.

5) It will inform the readers if the authors share the number of model updates (independent of the paired dataset size) used for PARP. One hypothesis for good performance could be the long training time the model weights are exposed to the paired data (without rewinding to their original pre-trained values). It is known that longer training times with slow learning rates lead to superior performance on many tasks.

**Time Spent Reviewing:**

2.5

---

> ### Author Response · Authors · 2021-08-10
> **Response to Reviewer xQYW (1/5)**
>
> We thank the reviewer for the constructive comments for PARP. We have done comprehensive additional experiments to improve the paper and broke down our responses to each of the five points raised by the reviewer.
>
> *Q1: The paper offers an imbalanced presentation between Speech and NLP. Given the generality of the proposed approach and continuous reference to previously proposed work on other modalities, it makes sense to offer more experiments on the same tasks. Adding Table 4 raises the confidence in the efficacy of the approach a bit. More experiments will convince the reader further.*
>
> The main setting for this work is on self-supervised models finetuned for low-resource ASR. We did not conduct extension NLP experiments since there is plenty of work on pruning BERT already [1-3], and our original intention was to target readers interested in speech and pruning. Section 4.5 and Table 4 were added mainly to compare to prior work BERT-Ticket on cross-task adaptation [4]. Section 4.6 was added to provide a grounding for our Observation 1, that though pretrained BERT/XLNet are heavily studied/dissected, it has not yet been the case for pretrained speech SSL. We do acknowledge the reviewer's point that we made heavy reference to prior NLP work in probing/pruning large pretrained models. We thus conduct the following additional extended NLP experiments:
>
> - BERT’s cross-task adaptation at additional sparsity levels (50%).
> - XLNet’s cross-task adaptation with baseline fineutning and our proposed PARP at 70% sparsity.
>
> Each set of experiments require 9*9 finetuning rounds given that there are 9 subtasks in GLUE. We first note that our NLP experimental designs are closely knitted to NLP probing work’s experimental setup [2,5], i.e. pretrained BERT/XLNet on 9 subtasks in GLUE. The experimental procedure is:
> - Finetune BERT/XLNet for a source task in GLUE.
> - Prune the finetuned model and obtain an OMP mask for each task.
> - Apply the OMP mask at pretrained initializations and finetune for a target task in GLUE w/ PARP.
>
> Below are the results averaged for each target task. As shown, we see PARP can mitigate cross-task adaptation in GLUE (c.f. same-task regular finetuning).
>
> - Averaged transferred subnetworks in BERT performance found at 50% sparsity
>
> Method  |  CoLA | MRPC | QNLI | QQP | RTE | SST-2 | STS-B | WNLI | MNLI
>  --- | --- | --- | --- | --- | --- | --- | --- | --- | --- |
> Same-task (top-line) | 50.46 | 83.58 | 90.88 | 90.92 | 62.45 | 91.97 | 88.04 | 56.34 | 84.15
> PARP | 56.02 | 84.23 | 90.76 | 91.07 | 67.51 | 92.76 | 89.33 | 54.46 | 84.15
>
> - Averaged transferred subnetworks in XLNet performance found at 70% sparsity
>
> Method  |  CoLA | MRPC | QNLI | QQP | RTE | SST-2 | STS-B | WNLI | MNLI
>  --- | --- | --- | --- | --- | --- | --- | --- | --- | --- |
> Same-task (top-line) | 29.92 | 76.47 | 89.62 | 90.74 | 59.21 | 92.2 | 80.78 | 42.25 | 85.16
> PARP | **30.09** | **77.56** | **87.10** | **90.66** | **58.88** | **91.73** | **83.80** | **52.11** | **83.87**
> BERT-Ticket | 11.47 | 74.16 | 85.21 | 89.11 | 55.80 | 90.19 | 75.61 | 42.25 | 82.65
>
> Secondly, we stress that what sets our work apart from the line of NLP probing/pruning work is that we are pruning the entire model weights, instead of only the contextualized representations at the output of a pre-selected layer. Therefore, it is not immediately clear whether findings in existing NLP work can smoothly be transferred to this setting.
>
> Reference: \
> [1] https://arxiv.org/pdf/1905.09418.pdf \
> [2] https://arxiv.org/pdf/2004.04010.pdf \
> [3] https://arxiv.org/pdf/2104.03514.pdf \
> [4] https://arxiv.org/pdf/2007.12223.pdf \
> [5] https://arxiv.org/pdf/2005.01172.pdf

---

> > ### Author Response · Authors · 2021-08-10
> > **Response to Reviewer xQYW (2/5)**
> >
> > *Q2: Using WER with no LM is accepted for seq2seq models which learn an implicit LM from the data; however, this work uses only a CTC loss leading to considerable variance in the WER when no LM is used, in particular when few amounts of paired data are used, e.g., 10mins. The authors are encouraged to share results with n-gram decoding, at least for their English models.*
> >
> > We first iterate the two reasons why we did not conduct LM decoding in the submission:
> >  - We would like to isolate the effect of pruning. Note that the standard LM (either 4-gram/transformer) used in the wav2vec series are also trained on Librispeech (text-corpus) [1,2]. Therefore, the LM can easily recover errors made by the acoustic model.
> >  - Secondly, note that the 4-gram/transformer LM decoding hyper-parameters are carefully searched via Bayesian optimization [3] in the wav2vec series. Such an optimization procedure would be quite expensive to run for **just one** model, let alone thousands of pruned models produced in this work.
> >
> > However, we conducted following experiments to address the reviewer’s two concerns raised here:
> > - *“considerable variance due to CTC loss with 10min data”* → we finetune wav2vec 2.0 base with 10min data at 10% sparsity with PARP at 8 additional seeds. Below is the result with Viterbi decoding w/o LM. We can see that at different seed values, models all converged to similar WERs, which is ~10% WER reductions compared to a full wav2vec 2.0 base.
> >
> > Model | test-clean/test-other (WER)
> > ----------- | ----------- |
> > Full wav2vec 2.0 | 49.3/53.2
> > wav2vec 2.0 + 10% PARP w/ seed 2447 | 38.04/44.33
> > wav2vec 2.0 + 10% PARP w/ seed 0 | 37.01/43.02
> > wav2vec 2.0 + 10% PARP w/ seed 1 | 37.82/43.66
> > wav2vec 2.0 + 10% PARP w/ seed 2 | 37.59/43.55
> > wav2vec 2.0 + 10% PARP w/ seed 3 | 37.57/43.29
> > wav2vec 2.0 + 10% PARP w/ seed 5 | 37.48/44.10
> > wav2vec 2.0 + 10% PARP w/ seed 6 | 37.87/43.55
> > wav2vec 2.0 + 10% PARP w/ seed 7 | 37.65/43.53
> > wav2vec 2.0 + 10% PARP w/ seed 8 | 38.22/43.91
> > wav2vec 2.0 + 10% PARP averaged | **37.69**/**43.66**
> >
> > - Below are the official 4-gram/transformer LM deciding results of our wav2vec 2.0 base finetuned with 10min data at 10% sparsity with PARP. Since we do not have the compute resource to replicate 1500 beam 4-gram decoding and 500 beam transformer LM decoding used in the original paper [1], we present results with a more reasonable beam size. The decoding hyper-parameters are searched via Ax on the dev-other Librispeech subset over 128 trials.
> >
> > Model | decoding algorithm | test-clean/test-other (WER)
> > ----------- | ----------- | ----------- |
> > Full wav2vec 2.0 | viterbi (no LM) | 49.3/53.2
> > Full wav2vec 2.0 | 4-gram LM + beam 5 | 27.82/**32.02**
> > Full wav2vec 2.0 | transformer LM + beam 5 | 27.16/32.68
> > wav2vec 2.0 + 10% PARP averaged | viterbi (no LM) | **37.69**/**43.66**
> > wav2vec 2.0 + 10% PARP averaged | 4-gram LM + beam 5 | **25.17**/32.13
> > wav2vec 2.0 + 10% PARP averaged | transformer LM + beam 5 | **25.45**/**32.46**
> >
> > As shown above, the performance gain over the full wav2vec 2.0 reduces with LM decoding, but we still observe a performance improvement at 10% sparsity with PARP.
> >
> > Reference: \
> > [1] https://arxiv.org/pdf/2006.11477.pdf \
> > [2] https://arxiv.org/pdf/2105.11084.pdf \
> > [3] https://github.com/facebook/Ax

---

> > > ### Author Response · Authors · 2021-08-10
> > > **Response to Reviewer xQYW (3/5)**
> > >
> > > *Q3: The results on initial subnetwork generality to multiple tasks for the speech case need to be taken cautiously since all the other tasks are still ASR (but in other languages). The bottom part of many models trained from waveform input captures spectral representations shared between languages for the same task. Audio Event Detection tasks or Speaker Clustering tasks may be good ones to examine this point for speech.*
> > >
> > > Our work focuses on low-resource ASR, given (1) ASR is still the focal point of most speech SSL work, see recent representative work [1-4], and (2) a standardized speech SSL benchmark, SUPERB [5], was only published right around the NeurIPS submission deadline.
> > >
> > > As a result, our finetuning experiments are all on ASR, and we did not overclaim Observation 1 to other speech downstream tasks. However, to investigate the hypothesis pointed out by the reviewer, we did a small-scale pilot cross-task adaptation experiment on SUPERB (cross-reference right of Figure 7). Note that none of the systems here are intended to achieve SOTA in each speech sub-task. The experiment procedure is as follows:
> > > - Take pretrained wav2vec 2.0 and finetune it on source subtasks in SUPERB.
> > > - Prune the finetuned models and obtain an OMP mask for each task.
> > > - Apply OMP mask back to pretrained wav2vec 2.0 and finetune for target subtasks in SUPERB with PARP.
> > >
> > > Here’re the results. Due to the time limit, we selected three target tasks as the finetuning criteria with source subnetworks coming from all tasks in SUPERB. The three target tasks are phone recognition with 10h Librispeech data, speaker verification on VoxCeleb, and Slot Filling on audio SNIPS. PER/EER/CER is lower the better, and F1 is higher the better.
> > >
> > > Source subnetwork from | target task 1: Phone-recog-10h (in PER) | target task 2: Speaker Verification (in EER) | target task 3: Slot FIlling (slot type F1/slot value CER)
> > > --- | --- | --- | --- |
> > > ASR-10h (50%) | 0.0567 | 0.1230 | 0.7635/0.4432
> > > ASR-1h (50%) |  0.0567  | 0.1316 | 0.7563/0.4470
> > > ASR-10min (50%) | 0.0576 | 0.1399 | 0.7452/0.4596
> > > Phone-recog-10h (50%) | 0.0471 | 0.1392 | 0.7575/0.4468
> > > Phone-recog-1h (50%) | 0.0483 | 0.1138 | 0.7508/0.4537
> > > Phone-recog-10min (50%) | 0.0535 | 0.1224 | 0.7519/0.4596
> > > Intent Class. (50%) | 0.0617 | 0.1165 | 0.7490/0.4621
> > > Keyword Spot. (50%) | 0.0656 | 0.1303 | 0.7490/0.4661
> > > Slot Filling (50%) | 0.0601 | 0.1097 | 0.7708/0.4327
> > > Speaker Verification (50%) | 0.0790 | 0.1131 | 0.7497/0.4654
> > > Speaker ID (50%) | 0.0677 | 0.1271 | 0.7581/0.4559
> > > Speaker Diarization (50%) | 0.0756 | 0.1104 | 0.7449/0.4623
> > >
> > > We first see that indeed the more similar source & target tasks are, the performance is better. For instance, source subnetwork obtained from speaker-related tasks perform better than those obtained from ASR/keyword spotting on speaker verification; source subnetwork obtained from ASR/phone recognition perform better than those obtained from speaker-related tasks on phone recognition. We do note that the results are not off by a lot, and the differences could be potentially reduced via hyper-parameter tuning. This pilot study also partially confirms with the reviewer’s intuition that subnetworks transferability depends on task similarity. Lastly, this experiment does not contradict our work’s setting, as we were primarily interested in cross-lingual transferability of subnetworks.
> > >
> > > Reference: \
> > > [1] https://arxiv.org/pdf/2105.11084.pdf \
> > > [2] https://arxiv.org/pdf/1812.09323.pdf \
> > > [3] https://arxiv.org/pdf/2006.11477.pdf \
> > > [4] https://arxiv.org/pdf/2006.13979.pdf \
> > > [5] https://arxiv.org/pdf/2105.01051.pdf

---

> > > > ### Author Response · Authors · 2021-08-10
> > > > **Response to Reviewer xQYW (4/5)**
> > > >
> > > > *Q4: There is not much discussion on where the pruned weights are located in the pre-trained network. The Wav2vec2.0 approach is known to have valuable representations towards the middle of the network, with higher-level weights only beneficial for the pre-training contrastive loss with less impact on downstream applications. One reason for the similarity between the subnetwork for English and multi-lingual models could be that both use the same pre-training approach, leading to the same inductive biases, i.e., a peculiarity of the pre-training approach rather than a property of the found sub-network used with PARP.*
> > > >
> > > > We separately respond to the two points raised here.
> > > > - We indeed do notice that intermediate **contextualized representations** contain valuable information for the downstream ASR task, as seen in recent work wav2vec-U [1], HuBERT [2], and SUPERB [3]. However, this piece of information does not contradict our Observation 1 since Observation 1 is made upon the **weights/neurons** of the whole network, instead of merely the per-layer contextualized representations. To further understand such a phenomenon though, we calculated the distributions of the pruned weights/neurons across each layer, shown below. We took a wav2vec 2.0 finetuned for Spanish (**H2L** setting) pruned at 50% sparsity with OMP.
> > > >
> > > > layer # | 1 | 2 | 3 | 4 | 5 | 6 | 7 | 8 | 9 | 10 | 11 | 12
> > > > -- | -- | -- | -- | -- | -- | -- | -- | -- | -- | -- | -- | -- |
> > > > sparisty (%) | 53.52 | 52.45 | 49.24 | 47.90 | 46.51 | 46.84 | 45.97 | 45.58 | 45.96 | 47.96 | 52.54 | 65.53
> > > >
> > > > We do observe similar pruned weight distributions across spoken languages (10 languages) and sparsities (10%, 20%, 30%, …). As predicted by the reviewer, the bottom and higher layers are all pruned more, while the middle layers are pruned less. This suggests that regardless of languages, intermediate layers’ neurons are more valuable than lower and higher-level layers. We will append this additional finding in the Appendix of the paper.
> > > >
> > > > - Regarding the latter point on the relation between Observation 1 and the inductive bias of wav2vec2/xlsr’s pretraining objective, we would like to first correct the reviewer’s statement *“a peculiarity of the pre-training approach rather than a property of the found sub-network used with PARP.”* Observation 1 has nothing to do with PARP. The “non-zero ASR pruning mask” in Observation 1 is obtained via the default pruning method OMP/IMP, generalized across pre-training model scale, spoken languages, and sparsities. In fact, Observation 1 forms the basis for PARP, not the other way around.
> > > >
> > > > Secondly, like the reviewer, we are also curious whether Observation 1 stands **across** pretraining methods; for instance, do wav2vec2’s non-zero ASR masks have high overlap with HuBERT’s non-zero ASR masks? Our pilot study during rebuttal indicates that while Observation 1 holds for wav2vec 2.0/HuBERT, it does not generalize **across** pre-training method. Below is the mask IOUs at 50% sparsity between wav2vec 2.0 base and HuBERT base on various speech subtasks in SUPERB (due to space limit, we only present wav2vec 2.0’s ASR-10h mask IOU with HuBERT’s speech subtasks’ masks):
> > > >
> > > > target task | mask IOU between wav2vec2 & hubert masks
> > > > -- | -- |
> > > > ASR-10h | 0.3472
> > > > ASR-1h | 0.3473
> > > > ASR-10min | 0.3473
> > > > Phone-recog-10h | 0.3473
> > > > Phone-recog-1h | 0.3473
> > > > Phone-recog-10min | 0.3473
> > > > Intent Class. | 0.3473
> > > > Keyword Spot. | 0.3473
> > > > Slot Filling | 0.3472
> > > > Speaker Verification | 0.3473
> > > > Speaker ID | 0.3473
> > > > Speaker Diarization | 0.3472
> > > > Random Pruning | 0.333
> > > >
> > > > The trend is the same across all 12 tasks in SUPERB, that wav2vec 2.0 (contrastive objective) and HuBERT (mask prediction objective) have close to random mask IOUs. Therefore, Observation 1 holds true **conditioned on** the same speech SSL pre-training objective. Note that for this set of experiments, we used the same optimization method (Adam w/ constant 1.0e-5 learning rate) for finetuning wav2vec 2.0 and HuBERT.
> > > >
> > > > Thirdly, this finding is perhaps not so surprising, given the prior work on similarity analysis between contextualized speech [4] and word [5] representations. They suggest that different pretrained models’ contextualized representations have low similarities, e.g. BERT v.s. XLNet. Lastly, we would like to stress again that this does not invalidate PARP. As long as Observation 1 holds, PARP’s step 2 should theoretically be able to make learnable adjustments to the initial mask given the high overlaps between pruning masks.
> > > >
> > > > References: \
> > > > [1] https://arxiv.org/pdf/2105.11084.pdf \
> > > > [2] https://ieeexplore.ieee.org/stamp/stamp.jsp?arnumber=9414460&casa_token=RzM0orKGtZEAAAAA:9o5WAfdf9FDXBL5n-iVmfc5COtCqamBs2FX_qqU2S-s54wf73PG4ytJ16t8v1Z_pBN09z9IKsQ \
> > > > [3] https://arxiv.org/pdf/2105.01051.pdf \
> > > > [4] https://arxiv.org/pdf/2010.11481.pdf \
> > > > [5] https://aclanthology.org/2020.acl-main.422.pdf \

---

> > > > > ### Author Response · Authors · 2021-08-10
> > > > > **Response to Reviewer xQYW (5/5)**
> > > > >
> > > > > *Q5: It will inform the readers if the authors share the number of model updates (independent of the paired dataset size) used for PARP. One hypothesis for good performance could be the long training time the model weights are exposed to the paired data (without rewinding to their original pre-trained values). It is known that longer training times with slow learning rates lead to superior performance on many tasks.*
> > > > >
> > > > > We already included the number of model updates for each finetuning setup in Appendix A.2, and we can move such information to the main paper if deemed necessary. Other hyper-parameters like learning rates/schedules are all the same as the standard wav2vec 2 model finetuning. Note that OMP/IMP have longer exposure to the data than PARP (Figure 1), yet with worse performance on nearly all conditions.

---

### Official Review · Reviewer_i9ds · 2021-08-05

**Rating:** 7
**Confidence:** 3

**Summary:**

The goal of this contribution is to reduce the complexity of self-supervised speech representation learning systems. Following a description of the landscape of unstructured magnitude pruning methods (i.e. methods that sort weights by their magnitude and remove the lowest ones with no other consideration), authors propose PARP. PARP is based on the observation that "task-agnostic" masks (obtained on a pre-trained network) are very similar to ones that would be obtained after fine-tuning on a target task. Taking benefit from this observation, and unlike OMP that cascades finetune -> prune -> finetune, PARP simply perform prune -> finetune. Experiments on low resource ASR and cross-lingual training show a clear advantage of PARP over IMP, OMP and MPI within a language (less obvious for high-to-low resource transfer and cross-lingual training), but also a surprising gain in performance w.r.t. unpruned architectures. Moreover, authors show that PARP also works on another modality, by pruning BERT models for GLUE downstream tasks.

**Main Review:**

The paper is remarkably well written and structured. The context and baselines are well explained, and as a reviewer unfamiliar with the pruning literature, I found the paper easy to follow and the models clearly explained. The experiments are convincing both in their scope (within and across languages, speech and NLP) and results. In particular, seeing improved performance over unpruned model with up to 50% sparsity in some cases is impressive. As a non-expert I cannot guarantee the appropriateness of the baselines (maybe some more sophisticated methods are missing), however given the baselines in the paper, PARP clearly has an advantage both in terms of training cost and performance. Overall, even though the problem addressed by this work could be considered as relatively narrow, the effort put in the redaction and presentation of the method and results, the significant performance of the system against baseline, as well as the generality of the approach (showcased by the NLP experiments) make this paper both worth reading and promising for future applications. For these reasons, I recommend this paper for acceptance.

Remarks:
* Figure 1 mentions an exponential increase in compute for IMP as the sparsity level increases. This is understandable following section 2.3 that explains how IMP prunes 10% of the remaining weights at each iteration (rather than 10% from all weights). However Figure 1 shows a linear increase in number of fine-tuning runs.
* Section 2.2 is in past tense, while the other sections are in present tense.
* line 151: "Initailizations"
* Algorithm 1: since m is updated at every iteration, it would be easier to read by adding an index to it.


**Time Spent Reviewing:**

4

---

> ### Author Response · Authors · 2021-08-10
> **Response to Reviewer i9ds**
>
> *Q: Figure 1 mentions an exponential increase in compute for IMP as the sparsity level increases. This is understandable following section 2.3 that explains how IMP prunes 10% of the remaining weights at each iteration (rather than 10% from all weights). However Figure 1 shows a linear increase in number of fine-tuning runs.*
>
> We thank the reviewer for the comments. We made a careless mistake in the wordings under Figure 1, and as pointed out, it should have been linear increase instead of exponential increase. Our intention for Figure 1 is to show that IMP requires much more compute cost than PARP for **each** downstream task/language, and such theoretical cost would become an issue when scaling up to, say 100+ languages.
>
> We will also correct the wordings/notations kindly pointed out by the reviewer.

---

### Comment · Reviewer_xQYW · 2021-08-12
**Update based on author feedback**

I'd like to thank the authors for their extensive experimental results added based on the review feedback. The authors are encouraged to add these findings/results in the main paper and/or the appendix. I'll raise my rating for the paper from 5 -> 7.

---

### Public Comment · ~Praveen_Narayanan1 · 2022-01-05
**Reasoning for better performance of pruned subnet**

This is fantastic work. One aspect that isn't apparent to me is why these pruned subnets should give better performance than the pretrained equivalent. Is it because overparameterization in the pretrained nets leads to overfitting (when data is limited)?

---

### Decision · Program_Chairs · 2021-09-27

**Decision:**

Accept (Spotlight)

**Comment:**

This paper is interested in reducing the complexity of self-supervised speech representation learning systems, by discovering sparse subnetworks in the self-supervised architecture which perform well on downstream tasks. While reviewers acknowledge the approach is simple (basically performing a sparsifying step followed by a finetuning one), it is effective. The paper also applies the idea on BERT models.

Authors should include in the final version of the paper the additional results and findings mentioned during the discussion period.